# Uveal melanoma immunogenomics predict immunotherapy resistance and susceptibility

Shravan Leonard-Murali[1,2,3,4], Chetana Bhaskarla [1,2,3], Ghanshyam S. Yadav[1,2,3], Sudeep K. Maurya[1,2,3], Chenna R. Galiveti[1,2,3], Joshua A. Tobin[1,2,3], Rachel J. Kann[1], Eishan Ashwat[1], Patrick S. Murphy[1,2], Anish B. Chakka [5], Vishal Soman[5], Paul G. Cantalupo[5], Xinming Zhuo[6], Gopi Vyas[6], Dara L. Kozak[6], Lindsey M. Kelly[6], Ed Smith [6], Uma R. Chandran[1,5], Yen-Michael S. Hsu[1,7,8] & Udai S. Kammula [1,2,3] ✉

Immune checkpoint inhibition has shown success in treating metastatic cutaneous melanoma but has limited efficacy against metastatic uveal melanoma, a rare variant arising from the immune privileged eye. To better understand this resistance, we comprehensively profile 100 human uveal melanoma metastases using clinicogenomics, transcriptomics, and tumor infiltrating lymphocyte potency assessment. We find that over half of these metastases harbor tumor infiltrating lymphocytes with potent autologous tumor specificity, despite low mutational burden and resistance to prior immunotherapies. However, we observe strikingly low intratumoral T cell receptor clonality within the tumor microenvironment even after prior immunotherapies. To harness these quiescent tumor infiltrating lymphocytes, we develop a transcriptomic biomarker to enable in vivo identification and ex vivo liberation to counter their growth suppression. Finally, we demonstrate that adoptive transfer of these transcriptomically selected tumor infiltrating lymphocytes can promote tumor immunity in patients with metastatic uveal melanoma when other immunotherapies are incapable.

Significant advances have been made in the treatment of metastatic cutaneous melanoma (CM) using immune checkpoint inhibition (ICI) targeting the cytotoxic T-lymphocyte-associated protein 4 (CTLA-4), the programmed cell death protein 1 (PD-1), and the lymphocyte-activation gene 3 protein (LAG-3)[1–5]. Unfortunately, ICI therapy has not shown comparable activity against most other solid tumors, especially those with low tumor mutational burden (TMB)[3,6]. To

improve immunotherapeutic strategies for this large group of unresponsive cancers we focused our studies on uveal melanoma (UM), a prototypic ICI resistant cancer with low TMB[4,7–10]. With an annual incidence of ~6 per million in Europe and the United States, UM is a rare cancer accounting for 3% of all melanomas[9,10]. Although both CM and UM develop from transformed melanocytes, UM uniquely arises from the pigmented epithelium of the uveal tract, an immune

[1]UPMC Hillman Cancer Center, University of Pittsburgh, Pittsburgh, PA, USA. [2]Solid Tumor Cellular Immunotherapy Program, UPMC Hillman Cancer Center, University of Pittsburgh, Pittsburgh, PA, USA. [3]Division of Surgical Oncology, Department of Surgery, University of Pittsburgh, Pittsburgh, PA, USA. [4]Department of Epidemiology, University of Pittsburgh, Pittsburgh, PA, USA. [5]Department of Biomedical Informatics, University of Pittsburgh, Pittsburgh, PA, USA. [6]UPMC Genome Center, University of Pittsburgh, Pittsburgh, PA, USA. [7]UPMC Immunologic Monitoring and Cellular Products Laboratory, University of Pittsburgh, Pittsburgh, PA, USA. [8]Division of Hematology/Oncology, Department of Medicine, University of Pittsburgh, Pittsburgh, PA, USA. ✉e-mail: kammulaus@upmc.edu

privileged site[11], and has an unusual predilection to aggressively metastasize to the liver which results in a dismal prognosis[9]. In further distinction, immunotherapies demonstrating efficacy against metastatic CM have shown disappointing results against UM[7,8], leading to speculation that UM is an immunologically 'cold' variant of melanoma. However, there has been recent therapeutic progress with the clinical introduction of tebentafusp, a bispecific glycoprotein 100 peptide-HLA-directed CD3 T cell engager, which has intriguingly improved overall survival in patients with metastatic UM yet has demonstrated only limited ability to mediate tumor regression[12,13]. To reconcile these paradoxical findings and develop more effective immunotherapeutics for metastatic UM, we sought to build upon our previous discovery that a subset of UM metastases naturally harbor tumor infiltrating lymphocytes (TIL) with potent autologous antitumor reactivity[14] and that adoptive cell therapy (ACT) administering such TIL could mediate cancer regression in 35% of patients with metastatic UM, including individuals who were refractory to ICI[15]. These observations suggested that occult immune responses exist and can be exploited to treat metastatic UM.

In this work we perform comprehensive immunogenomic profiling on the largest and most diverse group of human UM metastases compiled to date to uncover the tumor microenvironmental properties that underlie its occult immunogenicity and promote its resistance and susceptibility to different classes of immunotherapy. We find that over half of these metastases harbor TIL with potent autologous tumor specificity, despite having low tumor mutational burden and resistance to prior immunotherapies, including ICI and the bispecific T cell engager tebentafusp. These T cell infiltrated metastases display activated antigen presenting cells, chronic interferon signaling, and diverse T cell receptor repertoires. However, we observe strikingly low intratumoral T cell receptor clonality and transcriptionally non-proliferative TIL within the tumor microenvironment even after ICI and tebentafusp therapy, demonstrating that these immunotherapies were insufficient to induce proliferation of the tumor reactive TIL. To harness the therapeutic potential of these quiescent TIL, we develop rapid tumor transcriptomic profiling to enable their selective in vivo identification and ex vivo liberation to counter their growth suppression. We demonstrate that adoptive transfer of these transcriptomic selected TIL can promote tumor immunity in patients with metastatic UM when other immunotherapies are incapable.

## Results

### Clinicogenomic landscape of metastatic uveal melanoma
One hundred metastases were surgically procured from 84 UM patients as part of eligibility screening for TIL ACT clinical trials at the National Cancer Institute and the University of Pittsburgh Medical Center between 2013 and 2022 (NCT01814046 and NCT03467516)[15,16]. Resected metastases originated from 11 unique anatomic locations (Fig. 1a), with liver as the predominant procurement site (56%) (Fig. 1b). Patient demographics revealed a median age of 56 years (range = 17–78) and an even gender distribution (52% female, 48% male) (Supplementary Data 1). Patients had extensive metastatic disease burdens, with 95% having liver involvement, 75% having elevated LDH levels, and 71% with M1B or M1C stage (AJCC 8th edition) (Fig. 1c). Metastases were harvested from both treatment naïve patients (24%) and treatment refractory patients (76%). Notably, 46 patients received prior ICI therapy (anti-CTLA4 only = 3, anti-PD-1 only = 11, sequential therapy = 8, combination therapy = 24) and 9 patients received tebentafusp, of whom none showed objective response (Fig. 1c). Somatic mutational analysis of the metastases confirmed a low TMB (median = 0.64 mutations per megabase) with ubiquitous and mutually exclusive presence of established UM driver mutations (GNAQ, GNA11, CYSLTR2 or PLCB4) and frequent secondary alterations of BAP1 (62%) and SF3B1 (42%) (Fig. 1c and Supplementary Fig. 1a–e).

Somatic copy number alterations included chromosome 3 loss (46%) and 8q gain (85%) (Fig. 1c). No associations were found between TMB and cohort demographics (Supplementary Fig. 1f and Supplementary Data 2). In sum, clinicogenomic profiling established this patient cohort to be broadly representative of advanced UM and the procured metastases to have canonical UM driver alterations and low TMB.

### Unbiased tumor transcriptomics reveals T cell-inflamed uveal melanoma metastases
Current tumor biomarkers for immunotherapy susceptibility, such as TMB and PD-L1, are rarely used in metastatic UM due to the uniformly low expression of these markers in this melanoma variant[3,4,17]. Thus, we first sought to discover alternative immune prognostic metrics by interrogating the transcriptome of UM metastases using total RNA sequencing and unbiased computational profiling. Further, to facilitate a clinically relevant and minimally invasive biopsy approach for in situ tumor characterization, we restricted our analysis to a single random biopsy from each resected metastasis (~2 mm central core fragment from 93 metastases and ~500,000 cells post tumor dissociation from 7 metastases). Principal component analysis (PCA) revealed the majority of transcriptional variance among the metastases was restricted to PCs 1, 2, and 3 (variance contributed: 19%, 13%, 12% respectively) while the remaining PCs (4–100) each contributed ~5% or less variance (Supplementary Fig. 2a). Metastases were then mapped according to the three main PC coordinates (PCs 1, 2, and 3) (Supplementary Fig. 2b, c). To determine whether specific cellular pathways and processes were associated with specific PCs, we correlated PC coordinates (1, 2, and 3) with enrichment scores for each of the canonical hallmark gene sets from the Human Molecular Signatures Database (Supplementary Data 3)[18]. Unsupervised clustering (Euclidean distance) of the PC-gene set correlations (Spearman's rho) identified 4 discrete clusters (A, B, C, and D) with unique biologic motifs (Fig. 2a). Cluster A included cellular metabolism pathways (MYC TARGETS V1, MTORC1 SIGNALING, OXIDATIVE PHOSPHORYLATION), Cluster B included immune and inflammatory signaling pathways (INTERFERON ALPHA RESPONSE, INTERFERON GAMMA RESPONSE, ALLOGRAFT REJECTION, IL2 STAT5 SIGNALING), Cluster C included liver dominant physiologic pathways (BILE ACID METABOLISM, COAGULATION, CHOLESTEROL HOMEOSTASIS), and Cluster D included cellular signaling and division (WNT BETA CATENIN SIGNALING, MYC TARGETS V2, G2M CHECKPOINT). The individual metastatic samples were further clustered by their relative expression of each of the hallmark gene set clusters to reveal striking variability across the tumor cohort (Fig. 2b). Having identified transcriptomic differences among the metastases, we next sought to determine whether any of the three PCs independently correlated with the expression of the gene set clusters. Average Spearman's rank correlation coefficients (rho) for each of the gene set cluster enrichment scores (A, B, C, and D) were mapped against the individual PCs (1, 2, and 3) (Fig. 2c). We observed that cluster A (cellular metabolism) was strongly correlated with PC3 (mean rho = +0.76) but also weakly correlated with the negative aspect of PC1 (mean rho = −0.27). Cluster B (immune and inflammatory signaling) was exclusively correlated with the negative aspect of PC2 (rho = −0.32). Clusters C and D were not found to independently correlate with any of the three PCs. Given the independent association of PC2 with Cluster B immune pathways, we postulated that PC2 coordinate position was predominantly driven by intrinsic immune and inflammatory gene expression in these metastases. As support, when the enrichment scores for T cell activation gene sets were mapped onto three-dimensional PCA plots of the metastases, we observed that each gene set had a significant inverse relationship with the PC2 axis (INTERFERON GAMMA RESPONSE versus PC2: rho = −0.56, p = 2.52e-9; INTERFERON ALPHA RESPONSE versus PC2: rho = −0.56, p = 2.89e-9; ALLOGRAFT REJECTION versus PC2: rho = −0.49, p = 2.27e-7) (Fig. 2d and Supplementary Fig. 2d).

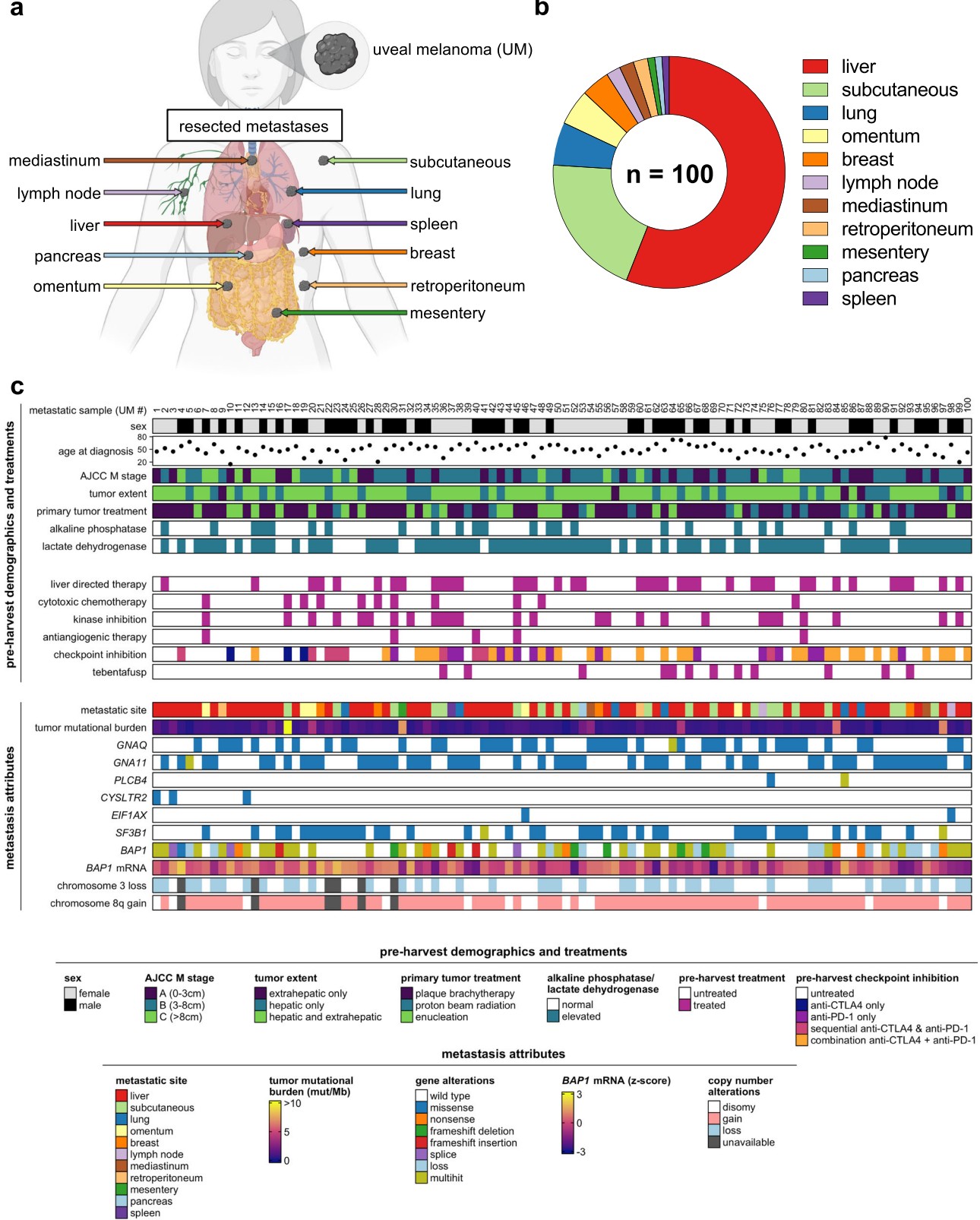

**Fig. 1 | Clinicogenomic landscape of metastatic uveal melanoma. a** Diversity of source tissues of resected metastases. Created with BioRender.com. **b** Distribution of source tissues of resected metastases. **c** Clinicogenomic annotation of individual metastases. Each column represents a single metastasis. *BAP1* mRNA *z*-scores were calculated using log$_2$(normalized counts).

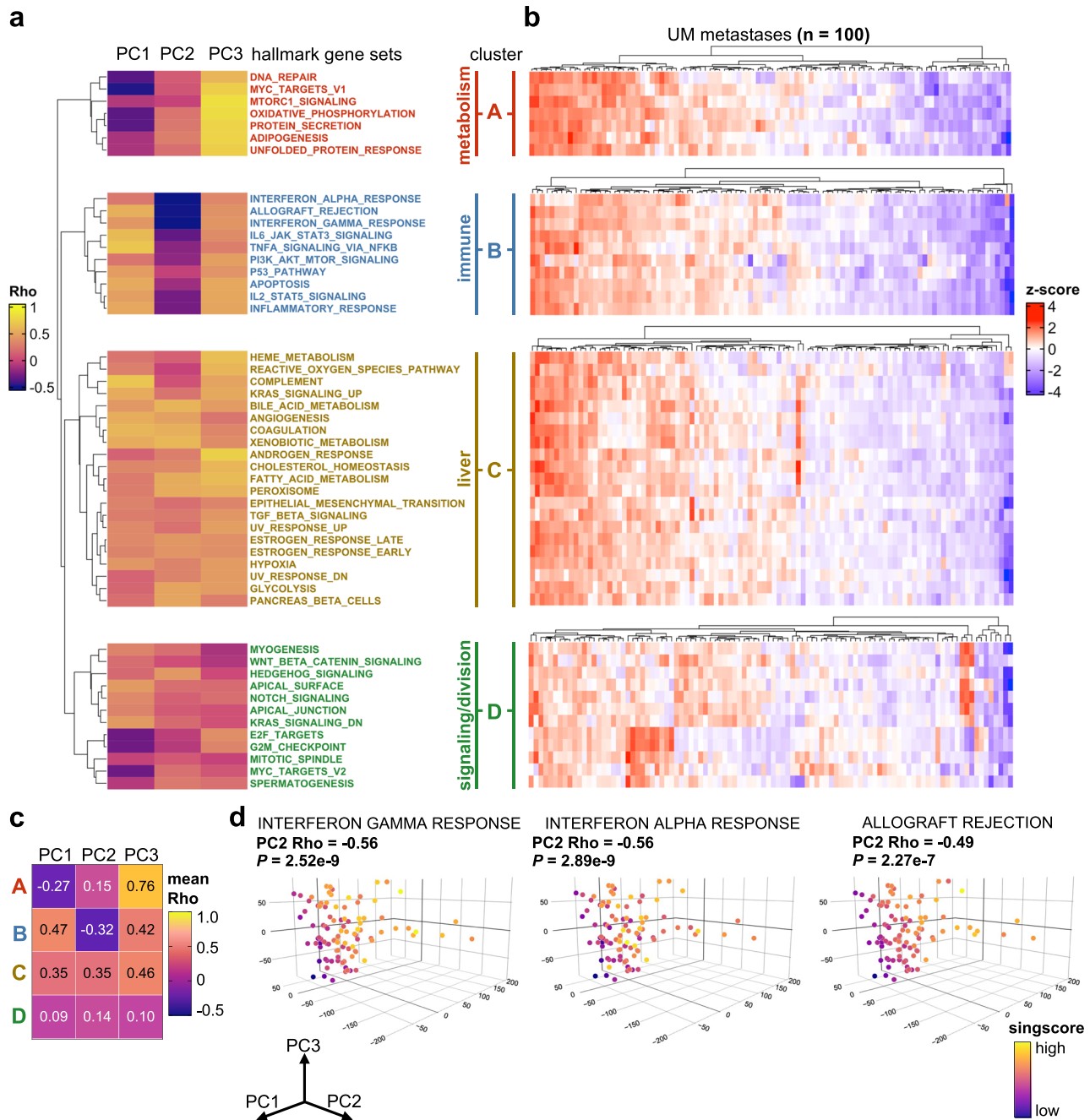

**Fig. 2 | Unbiased tumor transcriptomics reveals T cell-inflamed uveal melanoma metastases. a** Unsupervised clustering of Spearman's rank correlation coefficients derived from correlating PC coordinates (columns) with enrichment of hallmark signatures (row) for each individual metastatic sample (*n* = 100). Natural clusters identified by the row dendrogram are split, labeled (A, B, C, D), annotated, and color coded for visualization. **b** Heatmaps illustrating heterogeneity of hallmark signature enrichment across UM metastases (*n* = 100). Rows correspond to hallmark signatures listed in (**a**). Columns within each heatmap represent individual metastases. Each heatmap was clustered by metastases separately to display tumor heterogeneity within each hallmark cluster. *Z*-scores were calculated per row. **c** Matrix of mean Spearman's rank correlation coefficients for each cluster-PC combination. **d** Three-dimensional PCA plots displaying enrichment scores for selective hallmark immune related pathways identified in cluster B. Euclidean distance was used for hierarchical clustering (**a**, **b**). Statistical comparisons were performed using Spearman's rank correlation (**a**–**d**).

Collectively, unbiased computational profiling revealed PC2 coordinate mapping as an effective initial approach to segregate UM metastases with T cell-inflamed transcriptomic attributes.

## Development of Uveal Melanoma Immunogenomic Score (UMIS)

We next sought to refine the rudimentary PC2 variable into a more specific and clinically applicable immune metric for UM metastases.

First, we defined the 2394 genes that positively correlated with immune and inflammatory hallmark gene set enrichment (those with negative PC2 loadings). Rather than biasing this gene list with supervised filtering, we utilized the entire list of 2394 genes to facilitate discovery of novel biologic processes. Further, to enable single-sample prospective analysis, we employed a cohort-independent rank-based gene set scoring method (singscore[18]) to calculate enrichment scores for individual biopsies based upon

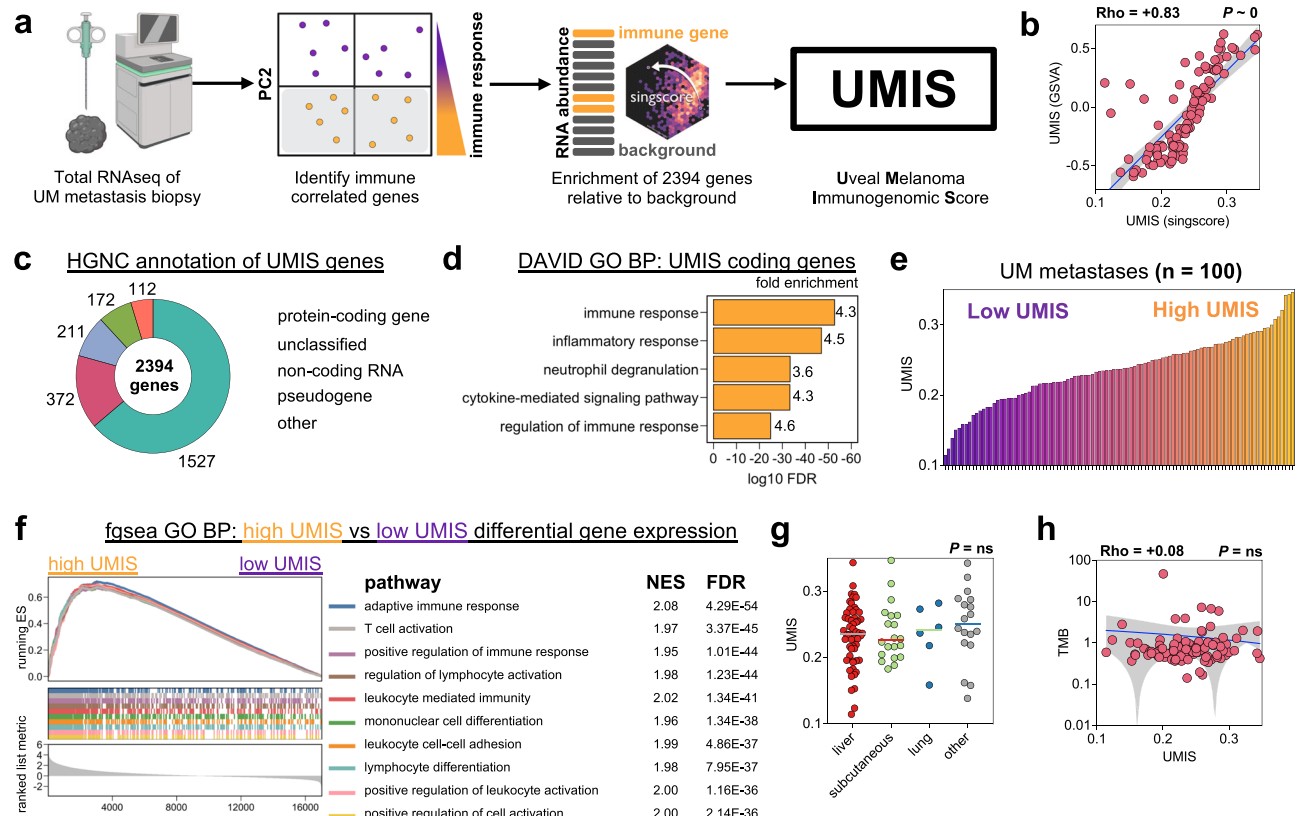

**Fig. 3 | Development of Uveal Melanoma Immunogenomic Score (UMIS).**
**a** Workflow for the development of UMIS. Created with BioRender.com.
**b** Correlation of UMIS scores calculated by the cohort-independent method, singscore, with UMIS scores calculated by the cohort-dependent method, gene set variation analysis (GSVA). **c** Annotation of UMIS genes using Human Genome Organization (HUGO) Gene Nomenclature Committee (HGNC). **d** Functional annotation of protein-coding genes within UMIS using Database for Annotation, Visualization and Integrated Discovery (DAVID) and Human Molecular Signatures Database Gene Ontology Biological Process gene set collection. **e** Distribution of

UMIS scores across the cohort of 100 metastases. **f** Gene set enrichment analysis of differentially expressed genes between high UMIS and low UMIS UM metastases. The ten pathways with the lowest FDR are displayed. **g** Comparison of UMIS by source tissue of resected metastases ($n = 100$ biologically independent samples; liver = 56, subcutaneous = 20, lung = 6, other = 18). **h** Correlation of UMIS with TMB. Statistical comparisons were performed using Spearman's rank correlation with overlaid simple linear regression to illustrate linearity (**b**, **h**), DAVID modified Fisher's exact test (**d**), fast preranked gene set enrichment analysis (**f**), or Kruskal–Wallis test by ranks (**g**).

transcript abundance (transcripts per million; TPM). Using this approach, we generated a single continuous variable for each metastasis called Uveal Melanoma Immunogenomic Score (UMIS) which reflected the concordance and mean percentile rank of our list of 2394 genes within the sample transcriptome (Fig. 3a). Cohort-independent UMIS (singscore) correlated strongly with the corresponding cohort-dependent score calculated using established pipelines (*GSVA*), supporting that our single-sample rank-based tool could be used for prospective evaluation of tumor biopsies without batch artifact (Fig. 3b)[18]. Of the 2394 genes that constitute UMIS, 1527 were protein-coding and the remaining 867 were a mixture of non-coding, unclassified, and pseudo genes (Fig. 3c and Supplementary Data 4). Functional annotation of UMIS coding genes confirmed pathways related to immune and inflammatory response (Fig. 3d). UMIS values ranged from 0.114 to 0.347 across the 100 metastases with a median score of 0.237, which was used as a cutoff to define high and low UMIS groups for categorical comparisons (Fig. 3e). Gene set enrichment analysis of high versus low UMIS metastases demonstrated that the most significantly enriched pathways were in the high UMIS group and involved T cell activation (Fig. 3f and Supplementary Fig. 3a). UMIS level was observed to be independent of metastatic site (Fig. 3g and Supplementary Data 5), TMB (Fig. 3h and Supplementary Data 4), somatic mutations and copy number alterations (Supplementary Fig. 3b and Supplementary Data 6 and 7), and class I human leukocyte antigen (HLA) alleles (Supplementary

Fig. 3c). Thus, UMIS represented a unique single-sample gene expression score derived from an unbiased mixture of coding, non-coding, and unannotated transcripts that could rank UM metastases based upon the expression level of immune and inflammatory genes.

## UMIS uncovers in vivo drivers of T cell recruitment and exclusion
To characterize the tumor microenvironmental cellular attributes contributing to UMIS, we performed whole-tumor single cell transcriptomics of six UM metastases with disparate UMIS values; 3 high UMIS (0.300, 0.268, 0.264) versus 3 low UMIS (0.199, 0.178, 0.162) (Supplementary Fig. 4a, b). We cataloged the 93,670 analyzed cells by building a unique metastatic UM single cell atlas using a two-step process that first categorized cells into large buckets (tumor, immune, and stroma) then assigned specific cellular and lineage labels (myeloid versus lymphoid) to the immune cellular fraction (Fig. 4a, b; Supplementary Fig. 4c–e and Supplementary Data 8 and 9)[19–21]. Our single cell analysis of low UMIS tumors had expectedly low numbers of immune cells. However, to maintain the true proportional landscape of specific cell types and avoid manipulation induced transcriptomic changes, we profiled the tumor digests without an additional enrichment step. We observed more lymphoid cells (proportion ratio = 10.50, $p = 0.047$) and fewer tumor cells (proportion ratio = 0.88, $p = 0.047$) in high UMIS versus low UMIS metastases (Fig. 4b and Supplementary Fig. 4f). Further, the composition of these lymphoid fractions differed, with the

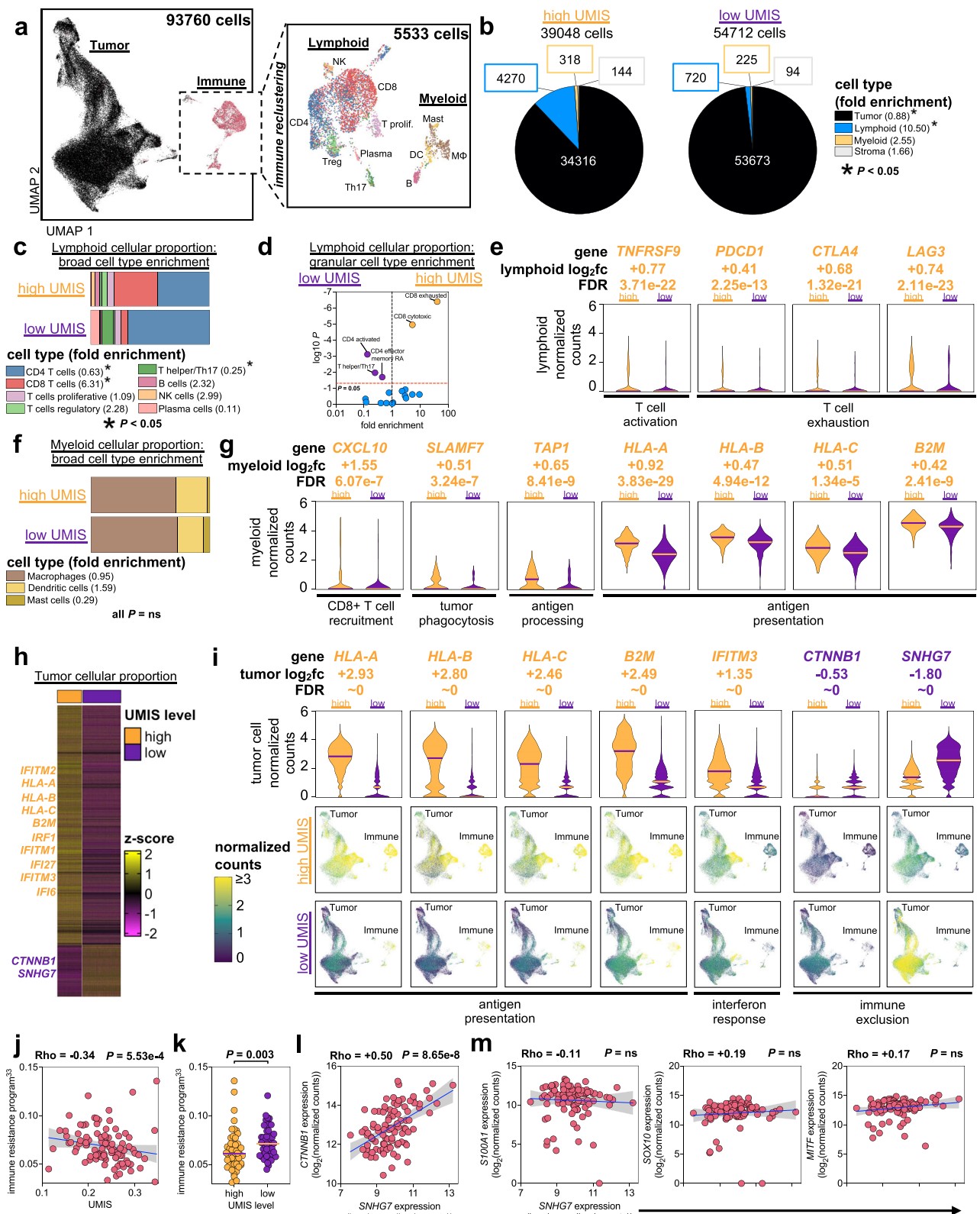

high UMIS metastases being enriched with CD8+ T cells (proportion ratio = 6.31, $p = 6.15e-7$) and the low UMIS metastases being enriched with CD4+ T cells (proportion ratio = 0.63, $p = 3.32e-4$) and T helper/ Th17 T cells (proportion ratio = 0.25, $p = 0.024$) (Fig. 4c). A granular analysis of the lymphoid cells revealed that the high UMIS metastases were enriched for CD8+ exhausted T cells (proportion ratio = 40.73, $p = 4.04e-7$) and CD8+ cytotoxic T cells (proportion ratio = 4.70,

$p = 1.13e-5$) (Fig. 4d). Intriguingly, we observed that 9% of the CD8+ exhausted and 20% of the CD8+ cytotoxic TIL retained transcriptomic expression of *TCF7*, suggesting possible progenitor capability (Supplementary Fig. 5a). Differential gene expression of the lymphoid cells revealed the high UMIS metastases had upregulation of genes involving T cell activation (*TNFRSF9, TNFRSF4*), T cell exhaustion (*PDCD1, CTLA4, LAG3, HAVCR2, VSIR*), lymphocyte activation (*STAT1*),

**Fig. 4 | UMIS uncovers in vivo drivers of T cell recruitment and exclusion.**
**a** Uniform manifold approximation and projection (UMAP) plot of all cells analyzed from 6 UM metastases. Magnified panel shows immune subset of cells after reclustering. Cell labeling is with a broad classification. **b** Proportion of overall cell types within UMIS groups. Fold enrichment refers to proportion ratio (high UMIS/ low UMIS). **c** Proportion of lymphoid broad cell types within UMIS groups. Fold enrichment refers to proportion ratio (high UMIS/low UMIS). **d** Volcano plot of lymphoid granular cell types within UMIS groups. Fold enrichment refers to proportion ratio (high UMIS/low UMIS). **e** Selected genes from differential gene expression analysis of high UMIS versus low UMIS lymphoid cells. Bars indicate medians and log$_2$fc refers to log2(fold change). **f** Proportion of myeloid broad cell types within UMIS groups. Fold enrichment refers to proportion ratio (high UMIS/ low UMIS). **g** Selected genes from differential gene expression analysis of high UMIS versus low UMIS myeloid cells. Bars indicate medians and log$_2$fc refers to log2(fold change). **h** Heatmap of differentially expressed genes between high UMIS and low UMIS tumor cells. Columns are individual cells, rows are genes. Cells are grouped by the UMIS level of their metastasis. The genes included had log2(fold change) ≥ |0.5| and FDR < 0.05. Z-scores were calculated per row. **i** Selected genes from differential gene expression analysis of high UMIS versus low UMIS tumor cells. Bars indicate medians and log$_2$fc refers to log2(fold change). UMAP plots display all cells within each UMIS subset. **j** Correlation of UMIS with immune resistance program scores[33] in UM metastases ($n = 100$). **k** Comparison of immune resistance program scores[33] by UMIS level in UM metastases (high UMIS $n = 50$, low UMIS $n = 50$; total $n = 100$ biologically independent samples). **l** Correlation of SNHG7 with CTNNB1 transcript expression in UM metastases ($n = 100$). Units are log2(normalized counts) from bulk RNAseq. **m** Correlation of SNHG7 with canonical melanoma marker transcripts (S100A1, SOX10, MITF) in UM metastases ($n = 100$). Units are log2(normalized counts) from bulk RNAseq. Statistical comparisons were performed using propeller (arcsin square root transformation of proportions) (**b–d, f**), Wilcoxon rank-sum test (two-tailed) (**e, g, i, k**) and Spearman's rank correlation with overlaid simple linear regression to illustrate linearity (**j, l, m**).

---

interferon response (*HLA-A, HLA-B, HLA-C, B2M, IFNGR1, IRF1, IFI27, IFI6, IFITM1, IFITM2, IFITM3*), T cell memory (*IL7R*), lymphocyte trafficking (*CXCL13, CCR7, SELL, CXCR3*), and T cell progenitor capability (*TCF7*) (Fig. 4e; Supplementary Fig. 5b and Supplementary Data 10)[22–28]. Taken together, these data demonstrate that the TIL found in high UMIS metastases had undergone activation and effector differentiation consistent with an in vivo adaptive anti-tumor response and indicative of a T cell-inflamed microenvironment.

We next investigated the myeloid cells found in high UMIS versus low UMIS metastases (Fig. 4f and Supplementary Fig. 5c). Although there was no enrichment of specific myeloid cell types (macrophages, dendritic cells, mast cells) in either group, differential gene expression revealed the high UMIS myeloid cells had upregulated genes involving CD8+ T cell recruitment (*CXCL10, CXCL9*), tumor phagocytosis (*SLAMF7*), antigen processing (*TAP1, TAP2*), antigen presentation (*HLA-A, HLA-B, HLA-C, B2M, HLA-DPB1, HLA-DQB1, HLA-DRB1*) and interferon response (*IRF1, IRF8, IFI27, IFI6*) (Fig. 4g; Supplementary Fig. 5d and Supplementary Data 11)[24,25,28–31]. These findings support that the T cell-inflamed microenvironment found in high UMIS metastases also included more active myeloid lineage antigen presenting cells (APCs) capable of recruiting CD8+ T cells.

Finally, since UMIS was derived using unbiased whole-tumor transcriptomics we postulated that this score may also reflect intrinsic differences among the tumor cells within high versus low UMIS metastases. Upon reclustering of tumor cells, we confirmed distinct separation of cells derived from high versus low UMIS metastases (Supplementary Fig. 6a). Differential gene expression revealed high UMIS tumor cells had significantly increased expression of several interferon-inducible transcription factors and elements (*IRF1, IFI27, IFI6, IFITM1, IFITM2, IFITM3*) and each of the major histocompatibility complex (MHC) class I molecule heterodimer components (*HLA-A, HLA-B, HLA-C, B2M*) (Fig. 4h, i; Supplementary Fig. 6b and Supplementary Data 12)[23–25]. These findings suggested that high UMIS metastases were composed of IFN-γ primed tumor cells that had upregulated MHC class I expression in response to chronic IFN-γ secretion from tumor specific CD8+ T cells[23,24]. In contrast, low UMIS tumor cells had 1.44-fold higher expression of *CTNNB1* (log$_2$(fold change) = −0.53, FDR ˜ 0) which encodes the beta-catenin protein (Fig. 4h, l and Supplementary Data 12). Activation of the Wnt/beta-catenin pathway has been implicated in T cell exclusion and may explain the paucity of CD8+ T cell infiltrate in low UMIS metastases[30,32]. In support, we found a significant inverse relationship across the total metastatic cohort ($n = 100$) between UMIS and the expression of a previously reported immune resistance program (Fig. 4j, k)[33]. Interestingly, the most upregulated gene in low UMIS tumor cells was the long non-coding RNA, *SNHG7*, which was 3.48-fold upregulated in low UMIS tumor cells (log$_2$(fold change) = −1.80, FDR ˜ 0) and has been previously reported as a positive regulator of *CTNNB1* expression in several cancers (Fig. 4h, i and Supplementary Data 12)[34–38]. Our findings confirmed a strong association between *SNHG7* and *CTNNB1* expression level in UM metastases ($n = 100$) (Fig. 4l) that was independent of tumor cell abundance as measured by melanoma-specific gene expression (*S100A1, SOX10, MITF*) and total RNA quantity (*ACTB, GAPDH*) (Fig. 4m and Supplementary Fig. 6c).

In sum, single cell transcriptomics demonstrated that UMIS was a holistic metric that reflected the gene expression of the lymphoid, myeloid, and tumor compartments within the tumor microenvironment. Further, UMIS classification of metastases revealed increased *CTNNB1* expression by low UMIS tumors cells as a putative driver of immune exclusion. In contrast, high UMIS metastases displayed lower tumor cell *CTNNB1* expression, more activated APCs, greater CD8+ T cell recruitment, and robust interferon signaling.

## UMIS predicts anti-tumor potency of ex vivo expanded TIL

To validate the transcriptomics demonstrating T-cell inflamed gene expression, we next interrogated the specific anti-tumor potency of the endogenous TIL from each of the UM metastases. Currently, the assessment of TIL tumor reactivity requires patients to undergo surgical resection of metastases followed by several weeks of ex vivo TIL expansion and finally resource intensive coculture with autologous tumor cells. Thus, we also investigated whether UMIS could serve as a rapid and minimally invasive clinical tool to predict the tumor specific potency of endogenous TIL. We compared UMIS values, derived from a single random biopsy from each source metastasis ($n = 100$), with the level of TIL anti-tumor reactivity found after conventional ex vivo expansion (Fig. 5a). TIL cultures ($n$ ˜ 24) were initiated from each freshly resected metastasis using a standardized ex vivo tumor fragmentation approach to address tumor heterogeneity, as previously described[15]. The individual TIL fragment cultures were tested for tumor specificity by coculture with autologous tumor digest (versus normal tissue controls) followed by measurement of 4-1BB upregulation on CD3+ cells (flow cytometry) and IFN-γ release (ELISA), which were found to be strongly correlated (Fig. 5b, c). The percentage of TIL cultures having tumor-specific reactivity from each metastasis was used as a standardized reactivity metric for comparing the level of anti-tumor TIL responses across tumors (Fig. 5d). We found the frequency of tumor reactive TIL cultures varied significantly among the total cohort (median = 6%; range = 0–100%) with 55 metastases having measurable anti-tumor reactivity and the remaining 45 metastases with no detected reactivity (Fig. 5e). Further, the metastases that had undergone prior ICI ($n = 53$) and tebentafusp therapy ($n = 12$) showed no difference in the mean percentage of tumor reactive TIL cultures when compared to samples that had not undergone these treatments (ICI 22% vs. no ICI 23%, $p$ = ns; tebentafusp 32% vs. no tebentafusp 21%, $p$ = ns) (Fig. 5e and Supplementary Data 13). The percentage of tumor reactive TIL cultures was also independent of metastatic site, TMB,

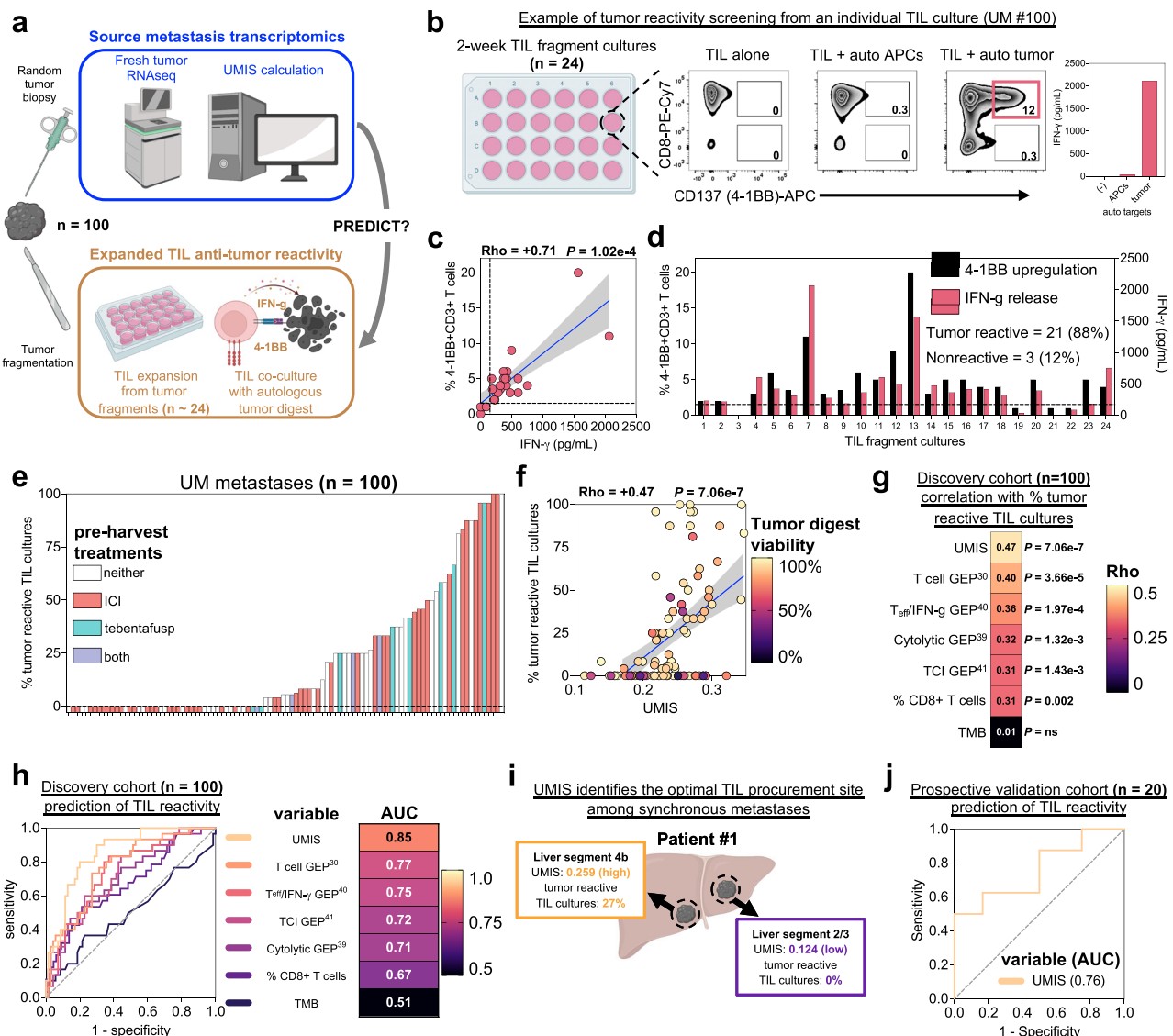

**Fig. 5 | UMIS predicts anti-tumor potency of ex vivo expanded TIL. a** Workflow for parallel analysis of source tumor transcriptomics and expanded TIL anti-tumor reactivity. Created with BioRender.com. **b** Example of TIL culture anti-tumor reactivity screening from source tumor UM #100. From left, tumor fragments (n = 24) are cultured individually for ~2 weeks before overnight coculture with autologous tumor cells and measurement of 4-1BB (CD137) expression by flow cytometry and IFN-γ release by ELISA. Final reactivity measurement subtracts background reactivity of TIL (TIL alone) and non-specific reactivity (TIL + autologous APCs). **c** Correlation between %4-1BB + CD3+ cells and IFN-γ release among the 24 fragment cultures from UM #100 after overnight tumor coculture. **d** Individual TIL fragment culture anti-tumor reactivity as assessed by 4-1BB upregulation and IFN-γ release from source tumor UM #100. TIL cultures were classified as tumor reactive if their 4-1BB expression was >1% (dotted line) and twice background or IFN-γ release was >100 pg/ml (dotted line) and twice background. Percentage tumor reactive TIL cultures was defined as 100*(tumor reactive TIL cultures)/(total TIL cultures). **e** Distribution of percent tumor reactive TIL cultures among the cohort of 100 metastases color-coded by pre-harvest treatments.

**f** Correlation of UMIS with percent tumor reactive TIL cultures. Color of each point denotes tumor digest viability percentage. **g** Correlative benchmarking of UMIS against published gene expression profiles and tumor biomarkers (n = 100 metastases). All correlations are with percent tumor reactive TIL cultures. **h** Predictive benchmarking of UMIS against published gene expression profiles and tumor biomarkers (n = 100 metastases). Receiver operating characteristic (ROC) curves and accompanying statistics are for variables' prediction of ≥33% tumor reactive TIL cultures. **i** Disparate UMIS and TIL culture reactivity from synchronous hepatic metastases in UM patient #1. Created with BioRender.com. **j** Validation of UMIS' ability to predict ex vivo TIL reactivity in an independent metastatic biopsy cohort (n = 20 metastases). Receiver operating characteristic (ROC) curve and area under curve (AUC) value is for UMIS' prediction of ≥33% tumor reactive TIL cultures. Statistical comparisons were performed using Spearman's rank correlation with overlaid simple linear regression to illustrate linearity (**c, f, g**) or univariate logistic regression (**h, j**). Gene expression profiles were calculated using singscore to best assess their cohort-independent predictive ability (Supplementary Data 16)[30,39–41].

specific mutation expression, copy number alterations, and class I HLA alleles (Supplementary Fig. 7a–d and Supplementary Data 13–15). When UMIS of each source metastasis was compared to the percentage of tumor reactive TIL cultures that were generated several weeks later, we found a strong positive correlation (rho = +0.47, p = 7.06e−7) (Fig. 5f). Notably, reactive TIL cultures were rarely expanded from metastases with a UMIS less than 0.2, suggesting the use of this cutoff

as a preoperative threshold to avoid futile surgical resection of non-inflamed UM metastases. Interestingly, we did observe a small subset of discordant metastases (n = 16) with high UMIS values that yielded TIL with no detectable anti-tumor reactivity based upon coculture with autologous tumor digest (Fig. 5f). However, upon assessing the quality of these specific tumor digest samples, we found they had significantly lower viability when compared to digests (n = 34) that yielded

concordance between high UMIS and co-culture reactivity (median digest viability: 78% vs. 93%, $p = 0.046$) (Fig. 5f and Supplementary Fig. 7e). Since UMIS quantitation was neither associated with nor dependent upon tumor digest viability (Supplementary Fig. 7f), the discordance with anti-tumor reactivity observed with this outlier subset likely stemmed from insufficient stimulatory capacity of these low viability tumor digests. To further characterize the performance of UMIS in our discovery cohort of 100 UM metastases, we benchmarked its ability to predict co-culture anti-tumor reactivity against several other tumor biopsy metrics including TMB, percentage of infiltrating CD8+ T cells, and several published gene expression profiles for T cell inflammation (Fig. 5g, h and Supplementary Data 16)[30,39–41]. We found that UMIS was the strongest performer as both a correlative metric (rho = +0.47, $p = 7.06e-7$) and classification metric (AUC = 0.85) for predicting ex vivo TIL reactivity (Fig. 5g, h). Not surprisingly, TMB had no predictive value (rho = +0.01, $p = ns$; AUC = 0.51) (Fig. 5g, h). In further support of UMIS as a preoperative biomarker, we found that UMIS level could identify metastases with the greatest yield of tumor reactive TIL among synchronous metastases in individual UM patients (Fig. 5i). In a prospective and independent validation cohort of metastatic UM biopsies, we corroborated the predictive ability of UMIS for ex vivo TIL reactivity ($n = 20$, AUC = 0.76) (Fig. 5j). Additionally, we validated that UMIS remained consistent across spatially distinct areas of individual tumors (Supplementary Fig. 8a) and could also be obtained from minimally invasive core biopsies (Supplementary Fig. 8b). Taken together, these findings establish that UMIS, obtained from a metastatic biopsy, could serve as a minimally invasive preoperative biomarker to both identify UM metastases harboring tumor reactive TIL and predict the percentage of tumor reactive TIL cultures that could be expanded without the limitations associated with conventional coculture assays.

## UMIS identifies quiescent TIL resistant to ICI and tebentafusp but sensitive to ex vivo expansion and adoptive transfer

Having found that UMIS strongly correlated with the level of TIL anti-tumor reactivity in metastases, we were surprised to find that high UMIS status was not significantly associated with improved survival in our UM cohort (Supplementary Fig. 9a). Furthermore, despite discovering tumor reactive TIL within the metastases of 23 UM patients (50%) who received prior ICI and 7 patients (78%) who received tebentafusp, we noted that none of these patients showed objective tumor regression with these therapies. To investigate these paradoxical findings, we analyzed the intratumoral TCR repertoire within the source metastases[42]. We found that the in situ diversity of the TCR beta (TRB; $n = 88$) and TCR alpha (TRA; $n = 82$) chains varied significantly across the total cohort of metastases (Shannon index ranges; TRB = 0.08–4.84, TRA = 0.69–4.72) (Fig. 6a and Supplementary Fig. 10a). UMIS was found to strongly correlate with both TRB diversity (rho = +0.54, $p = 5.40e-8$) (Fig. 6a) and TRA diversity (rho = +0.45, $p = 2.14e-5$) (Supplementary Fig. 10a) suggesting that high UMIS metastases had more polyclonal T cell infiltrates. In contrast, TRB and TRA clonality, an in vivo surrogate for relative TIL clonal expansion, was low and minimally variant across the samples (1 – Pielou's index ranges; TRB = 0–0.89, TRA = 0–0.23) (Fig. 6a and Supplementary Fig. 10a). We expectedly found no correlations between UMIS and the clonality of the TRB (rho = +0.02, $p = ns$) (Fig. 6a) and TRA (rho = −0.07, $p = ns$) chains (Supplementary Fig. 10a). The in vivo quiescence of these TIL was further corroborated by single cell TCR repertoire analysis demonstrating low clonality (Supplementary Fig. 10b) and single cell transcriptomics which found that the percentage of proliferative T cells was equivalently low in high and low UMIS metastases (Figs. 4c and 6b). Interestingly, prior ICI therapies ($n = 53$) had no influence on TCR diversity compared with untreated samples (Fig. 6a, c and Supplementary Fig. 10a, d). In contrast, prior tebentafusp treatment ($n = 12$) was associated with greater TCR diversity, consistent with the

ability of this bispecific T cell engager to recruit T cells to these metastases (Fig. 6a, c and Supplementary Fig. 10a, c, d). However, neither prior ICI nor tebentafusp therapy were associated with an increase in TCR clonality (Fig. 6a, c and Supplementary Fig. 10a, c, d), indicating that these immunotherapies were incapable of inducing in vivo proliferation of the endogenous TIL. When specific TRB and TRA sequences were compared across the metastases ($n = 100$), we found that most of the sequences were private, with rare and limited public expression suggesting unique, rather than shared, antigen targeting (Supplementary Fig. 10e). Cumulatively, these TCR repertoire studies demonstrated that although high UMIS metastases were infiltrated with a unique polyclonal population of TIL, these T cells remained quiescent, even after receiving ICI and tebentafusp therapy.

To determine whether the deficient proliferation of the intratumoral T cells was due to T cell exhaustion or other intrinsic proliferative defects, we performed clinical scale ex vivo rapid expansion (REP) of TIL from UM metastases that were either naïve to ICI and tebentafusp ($n = 3$), or refractory to ICI ($n = 10$), tebentafusp ($n = 4$), or both therapies ($n = 2$) (Fig. 6d, e and Supplementary Fig. 11a–c). We observed that TIL from each of the metastases demonstrated approximately 5-log expansion, reaching massive cell counts (median = 7.31e10, range = 1.00e10–1.12e11) (Fig. 6d, e and Supplementary Fig. 11a). Further, these expanded TIL demonstrated a significant decrease in TCR diversity ($p = 4e-6$) and a significant increase in TCR clonality ($p = 4e-6$) as compared to their source metastases using highly specific targeted TCR sequencing (Fig. 6d, e and Supplementary Fig. 11a–c). These findings suggested that the endogenous TIL were not limited by intrinsic proliferative deficiencies, but instead their growth was likely suppressed by the tumor microenvironment. Taken together, we observed the quiescence of endogenous TIL in UM metastases was not reversed with ICI or tebentafusp but could be revived with ex vivo liberation and expansion.

Based upon the observation that UMIS from a metastatic biopsy could predict the ex vivo potency of quiescent endogenous TIL, we postulated that UMIS might also predict the clinical efficacy of adoptive transfer of these TIL after ex vivo liberation and expansion (Fig. 6f). Of the 100 UM metastases profiled, 19 had been used to manufacture TIL for a previously reported ACT trial in patients with metastatic UM (NCT01814046)[15]. Among this treatment cohort, which included 6 responders and 13 nonresponders, we observed a strong correlation between source tumor UMIS and the ex vivo anti-tumor reactivity of the post-REP TIL infusion product ($n = 17$, rho = +0.61, $p = 0.011$; 2 infusion products were not tested due to insufficient tumor) (Fig. 6g). Additionally, we found that UMIS as a continuous variable strongly correlated with magnitude of clinical tumor regression after adoptive transfer in patients with metastatic UM, including ICI refractory individuals ($n = 19$, rho = −0.68, $p = 0.001$) (Fig. 6h). To help define a UMIS threshold value that might have future clinical utility in predicting RECIST objective responses (≥30% reduction), we utilized the median UMIS value of the non-responder group (UMIS = 0.246) as a response threshold (Fig. 6i). We observed that patients having source metastases above this threshold had significantly improved progression-free and overall survival after TIL ACT versus those below the threshold (Fig. 6j). In sum, these findings demonstrate that UMIS, performed on a pre-treatment metastatic biopsy, correlated with the clinical outcome after adoptive transfer of TIL and may serve as a future predictive biomarker for the treatment of metastatic UM with ACT.

## Discussion

Here we profiled the immunogenomic landscape of metastatic UM using bulk and single cell transcriptomics, TCR repertoire analysis, TIL reactivity assessment, and clinical adoptive cell therapy. Our findings establish that metastatic UM is not an immunologically 'cold' cancer, but instead over half of the analyzed UM metastases harbored tumor reactive TIL, despite having one of the lowest mutational

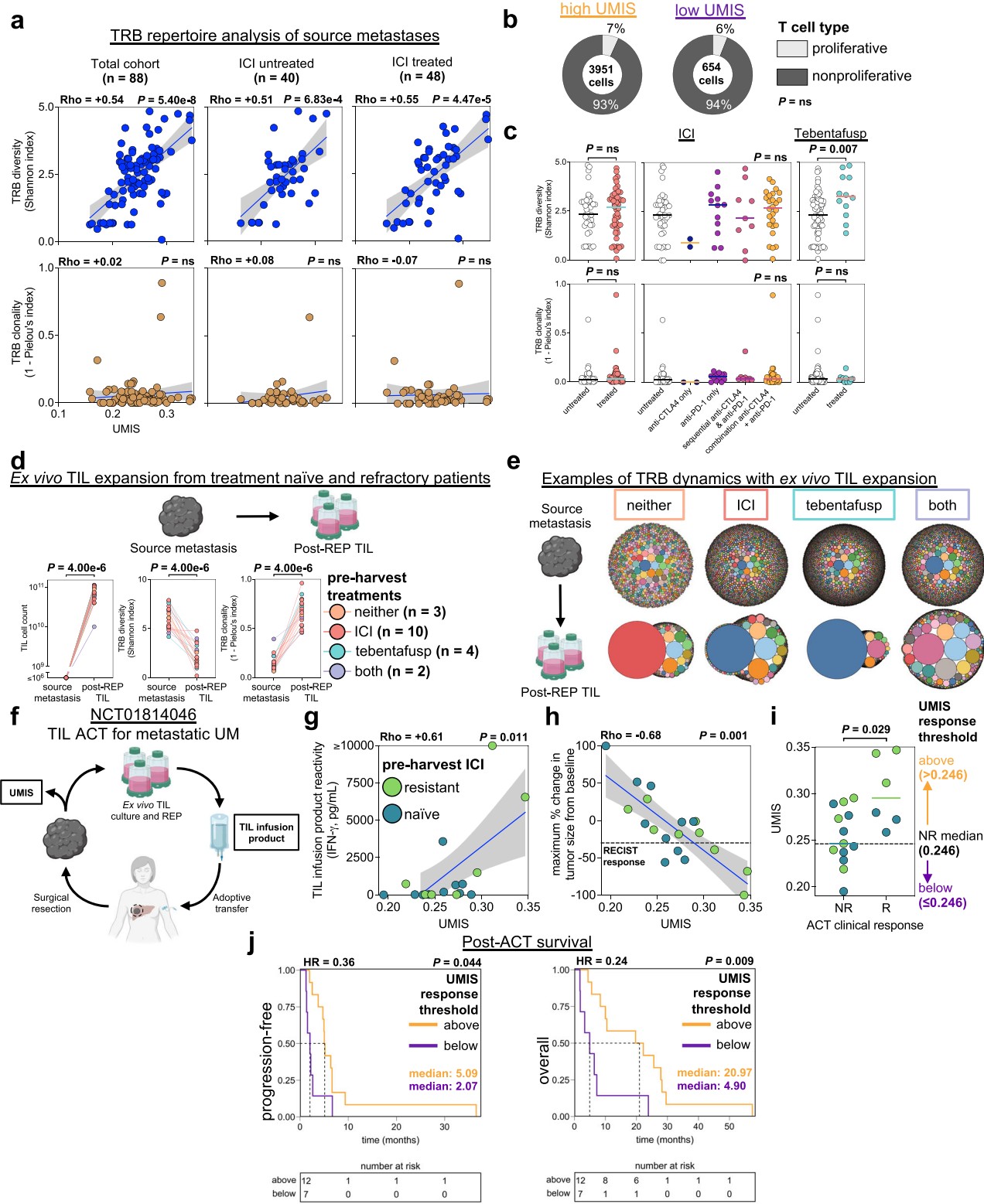

burdens of any solid cancer[4,9,43,44] and an equally limited responsiveness to approved immunotherapies, including ICI therapy and tebentafusp[4,7,8,12,13,45,46]. To avoid sampling bias in our study, we analyzed a large clinically representative group of UM patients ($n = 84$) and their metastases ($n = 100$) which were procured from a diverse array of organ sites ($n = 11$). Further, the metastases in the current study were genomically validated to be of uveal origin by expression of canonical UM somatic alterations and low TMB[4,9,43,47,48]. Thus, we believe our study cohort to accurately represent the metastatic immunogenomic

landscape of this rare cancer and uniquely suited to answer two critical questions that have significant therapeutic implications for UM: what factors drive T cell inflammation in metastatic UM, and why does metastatic UM respond so poorly to currently approved immunotherapies?

To define drivers of immune response against metastatic UM, we utilized bulk total RNA sequencing of metastatic biopsies and found that T cell-inflamed metastases naturally segregated from T cell excluded metastases based upon an unsupervised transcriptomic signature

**Fig. 6 | UMIS identifies quiescent TIL resistant to ICI and tebentafusp but sensitive to ex vivo expansion and adoptive transfer. a** T cell receptor beta (TRB) repertoire analysis of bulk RNAseq of UM metastases ($n = 88$). Immune checkpoint inhibition (ICI) refers to treatment history prior to metastatic biopsy. **b** Proportion of proliferative T cells in UMIS groups by single cell atlas. **c** Comparisons of TRB diversity and clonality in ICI or tebentafusp untreated versus treated metastases (ICI: 48 treated, 40 untreated; tebentafusp: 12 treated, 76 untreated). **d** Ex vivo TIL expansion from treatment naïve and refractory patients ($n = 19$). Listed therapies were received prior to metastatic biopsy. Changes in TIL cell counts (left), T cell receptor beta (TRB) diversity (middle), and TRB clonality (right) are shown for source metastases and corresponding TIL cultures post rapid expansion protocol (post-REP TIL). Metastases' TIL counts were conservatively estimated to be ≤$10^6$. TRB repertoires were characterized with targeted TCR repertoire analysis. Schematic created with BioRender.com. **e** Examples of TRB dynamics with ex vivo TIL expansion. Bubble plots represent unique TRB clonotypes (color coded) with bubble size indicating percentage of total clonotypes. Shown are representative examples for each pre-harvest treatment group (neither = UM #73, ICI = UM #50,

tebentafusp = UM #59, both = UM #49). Schematic created with BioRender.com. **f** Schematic for evaluation of UMIS in the context of NCT01814046 (ACT of TIL for metastatic UM)[15]. Created with BioRender.com. **g** Correlation of source metastasis UMIS with TIL infusion product reactivity ($n = 17$). **h** Correlation of source metastasis UMIS with maximum percent change in tumor size from baseline (RECIST v1.1) after TIL ACT. RECIST response line is drawn at −30% ($n = 19$). **i** Comparison of UMIS between responders (R; $n = 6$) and nonresponders (NR; $n = 13$) to TIL ACT. The median UMIS of the NR group (0.246) was used as a clinical response threshold for outcome analyses. **j** Time-to-event curves of post-ACT survivals by UMIS response thresholds ($n = 19$). Progression-free survival used progressive disease as the event (median follow-up (months): high = 5.09, low = 2.07). Overall survival used death as the event (median follow-up (months): high = 20.97, low = 4.90). Hazard ratios (HR) are for above versus below threshold groups. Statistical comparisons were performed using Spearman's rank correlation with overlaid simple linear regression to illustrate linearity (**a**, **g**, **h**), Fishers exact test (**b**), Wilcoxon rank-sum test (two-tailed) (**c**, **i**), Kruskal–Wallis test by ranks (**c**), Wilcoxon signed-rank test (two-tailed) (**d**) or logrank test (**j**).

---

composed of coding, non-coding, and unannotated transcripts. Rather than biasing this gene list with supervised filtering, we integrated the entire 2394 gene set into a gene expression score called UMIS. Based upon a unique single cell transcriptomic atlas that we developed specifically for metastatic UM, we found that UMIS could holistically reflect the multiple cellular components of the tumor microenvironment[49]. Metastases with low UMIS (versus high UMIS) had a paucity of TIL and were composed of tumor cells with higher beta-catenin transcript expression (*CTNNB1*), which has been described as a transcriptional repressor of *BATF3*-lineage dendritic cell recruitment of CD8+ T cells[30,50,51]. In contrast, high UMIS metastases had lower tumor cell expression of *CTNNB1*, increased APC expression of T cell chemoattractant ligands (*CXCL10* and *CXCL9*), greater tumor reactive TIL recruitment, and markedly elevated MHC expression on multiple cell populations within the tumor microenvironment, suggesting prominent interferon signaling. Thus, we believe Wnt/beta-catenin signaling to play an important role in promoting immune exclusion in metastatic UM, similar to prior reports in metastatic CM[30,32,50–52]. Surprisingly, we identified only a single metastasis with a possible activating somatic mutation of the Wnt/beta-catenin pathway, suggesting hotspot mutations are not a common driver of beta-catenin overexpression in UM metastases[53]. However, we did observe a strong correlation between the expression of the long non-coding RNA, *SNHG7*, and *CTNNB1*. Based upon several reports that *SNHG7* is a positive regulator of *CTNNB1* and compelling evidence that in vitro knockdown of *SNHG7* leads to downregulation of the Wnt/beta-catenin pathway in various other cancers[34–38] we are investigating the mechanistic role of this non-coding RNA in driving T cell exclusion in UM metastases and therapeutic strategies to potentially abrogate its effect in low UMIS metastases.

To better understand why UM metastases rarely regress with approved immunotherapies, we evaluated TIL from ICI and tebentafusp resistant patients. From our total cohort of TIL samples ($n = 100$), we observed that 55% of UM metastases harbored tumor reactive TIL and there was no difference in the percentage of tumor reactive TIL cultures expanded from ICI and tebentafusp treated metastases versus untreated. Yet, despite the presence of potent TIL in these metastases, we found they were strikingly quiescent with an absence of in vivo TIL expansion using TCR clonality analysis and single cell transcriptomics. Interestingly, we observed prior tebentafusp therapy was associated with increased in vivo TCR diversity in our samples, demonstrating its ability as a T cell recruiter[12,13]. However, neither tebentafusp nor ICI therapy were associated with an increase in TCR clonality. The quiescence of these T cells within the tumor microenvironment may explain the low rates of objective tumor regression and the intriguing decoupling of overall response rate as a surrogate for overall survival in UM patients treated with tebentafusp[12,13,54]. In contrast, TCR repertoire studies of cutaneous melanoma metastases have reported significant

variance in TCR clonality, with higher pre-treatment clonality being associated with improved response to PD-1 blockade[55–58]. Interestingly, we found the quiescent TIL from ICI and tebentafusp treated UM metastases could demonstrate significant ex vivo expansion, indicating that these T cells were not limited by intrinsic factors such as exhaustion, but rather by extrinsic constraints within the UM tumor microenvironment. In support, we previously observed that adoptive transfer of tumor reactive TIL could mediate objective regression in ICI refractory UM patients[15]. In sum, these findings reveal that occult T cell responses do exist against metastatic UM but require therapeutic strategies such as ACT to overcome their growth-suppressed state within the tumor microenvironment.

Finally, our study revealed the importance of UMIS as a tumor intrinsic biomarker to predict TIL potency and clinical response after adoptive transfer in UM patients. Whereas recent reports have proposed phenotypic and transcriptomic markers for the purpose of defining neo-antigen specific TCR sequences from TIL[27,59,60], we believe UMIS represents a unique tumor biomarker for the identification of tumor reactive TIL capable of ex vivo expansion for clinical adoptive transfer. Importantly, a UMIS level of less than 0.2 identified metastases that were unlikely to yield potent TIL, suggesting that pre-operative UMIS measurement could prevent futile invasive surgical harvests. We found UMIS performed significantly better as a tumor intrinsic biomarker of TIL potency when compared to several focused gene expression signatures of T cell inflammation. We postulate that the superior performance of UMIS was a result of its unique derivation from a large unbiased mixture of coding and non-coding transcripts. Further, rather than narrowly reflecting the gene expression of only immune cells, UMIS was developed as a whole-tumor metric that reflected the gene expression of the lymphoid, myeloid, and tumor compartments within the tumor microenvironment.

Potential limitations of our study include selection bias of the patients and metastases analyzed. The rare nature of UM limited our sample size to 100 metastases. A subset of patients contributed multiple metastases to the study (14 patients with multiple metastases included). We chose to include these metastases to maximize sample size in this rare cancer but recognize that this may be a source of selection bias. Given that UM patients presented with the intent of ACT screening, which has strict eligibility requirements, the analyzed cohort may not represent elderly individuals or those demonstrating rapidly progressive metastatic disease and declining performance. Further, although UMIS was found to correlate with TIL reactivity across 24 geographically unique tumor fragments, the predictive ability of UMIS in larger and more heterogenous lesions needs further study. Finally, while our ongoing independent validation cohort ($n = 20$) has corroborated the ability of UMIS to predict TIL reactivity, longer clinical follow up is required to evaluate survival in this group of patients.

## Methods

### Patient samples and clinical annotation

Patients were screened and tumor samples were obtained after informed consent in conjunction with tumor procurement banking protocols associated with two adoptive TIL transfer clinical trials: NCT03467516 (Hillman Cancer Center, UPMC, Pittsburgh, PA, USA) and NCT01814046 (Surgery Branch, NCI, Bethesda, MD, USA). Patients gave informed consent in accordance with the Declaration of Helsinki, and the trials were reviewed and approved by the NCI and UPMC Institutional Review Boards. There was no requirement for previous systemic therapy, given the lack of known effective systemic treatments for metastatic UM at the time of study. If patients did receive previous systemic treatment, more than 4 weeks must have elapsed before initiation of the current trial therapy, and patients' toxicities must have recovered to a grade 1 or less (except for toxicities such as alopecia or vitiligo). All patients were required to have progressive and measurable metastatic disease with an Eastern Cooperative Oncology Group performance status of 0 or 1 and life expectancy greater than 3 months at the time of enrollment. Patients were required to have adequate hematological, renal, and hepatic function. Patients were excluded if they had active systemic infections, coagulation disorders, or other active major medical illnesses of the immune system[15].

Clinical information, including demographics and treatments, were collected relative to the date of metastatic tumor harvest. Sex was self-reported by patients and consistent with biological sex determined by genomic analysis. Time-to-event data was collected relative to multiple dates: date of primary diagnosis, date of metastatic diagnosis, and date of ACT (if applicable). In cases of patients with multiple metastatic biopsies an algorithm was adopted for selection of a representative biopsy for the purposes of patient-centered time-to-event analysis (in descending order of priority: metastasis harvested prior to any ACT, metastasis whose TIL was utilized for subsequent ACT, more recently harvested metastasis. Response to ACT was evaluated using RECIST v1.1 criteria[61].

### Tumor procurement, ex vivo TIL culture and tumor reactivity testing

All patients had surgical metastatectomies as screening for clinical trials NCT03467516 (Hillman Cancer Center, UPMC, Pittsburgh, PA, USA) and NCT01814046 (Surgery Branch, NCI, Bethesda, MD, USA) to procure tumor tissue to generate autologous TIL for therapy[16]. After surgical procurement of a metastatic lesion, the fresh tumor underwent sterile dissection in the UPMC Immunological and Cell Products Laboratory (Pittsburgh, PA, USA) or the Surgery Branch Cell Production Facility (Bethesda, MD, USA). Representative samples of tumor were sent for formal pathological confirmation of UM. Tumor samples procured at the Surgery Branch, NCI, were transferred via a Material Transfer Agreement to the University of Pittsburgh where the analyses were conducted. To develop a clinically relevant core biopsy approach for in situ tumor characterization, a single random biopsy was obtained from each resected metastasis (~2 mm central core fragment from 93 metastases and ~500,000 cells post tumor dissociation from 7 metastases) and were snap-frozen in liquid nitrogen and stored long term at −80 °C for future DNA and RNA extraction.

TIL cultures were initiated from geographically discrete 1–2 mm³ tumor fragments ($n$ - 24) that were placed individually in wells of a 24-well culture plate containing complete media with human AB serum and recombinant interleukin-2 (6000 IU/ml; Clinigen). Remaining fresh tumor was processed by mechanical and enzymatic digestion with the human Tumor Dissociation Kit (Miltenyi Biotec) and gentle-MACS Dissociator (Miltenyi Biotec) to provide a single cell suspension of autologous tumor targets for TIL reactivity testing. Tumor digests underwent flow cytometric phenotyping with propidium iodide followed by the following anti-human monoclonal antibodies: CD3-APC-Cy7, CD8-PE-Cy7, CD4-PE (BD Biosciences). Tumor digest viability was

determined by percent propidium iodide negative cells by flow cytometry or percent trypan blue negative cells by manual cell counting (UM #47). After ~2 weeks of growth, individual TIL fragment cultures were tested for tumor specificity by coculture with autologous tumor cells (versus normal tissue controls) followed by measurement of 4-1BB upregulation on CD3+ cells by flow cytometry using anti-human CD137 (4-1BB)-APC (BD Biosciences) and IFN-γ release by ELISA[14,15]. Tumor single cell suspensions (digests) and peripheral blood mononuclear cells were cryopreserved in freezing media and stored long term in liquid nitrogen. Monocytes were isolated from peripheral blood mononuclear cells using the CD14+ MicroBead isolation kit (Miltenyi Biotec). All flow cytometry data was analyzed with FlowJo v10.8 Software (BD Life Sciences).

Clinical scale ex vivo rapid expansion protocol (REP) involved selection of individual fragment T cell cultures for further expansion based on proliferative capacity and evidence of autologous tumor reactivity. Final large-scale expansion of selected TIL cultures was done with anti-CD3 antibody (30 ng/ml, Ortho Biotech or 50 ng/ml, Miltenyi Biotec) and recombinant interleukin-2 (3000 IU/ml; Clinigen) in the presence of irradiated peripheral blood mononuclear feeder cells[15]. The specific anti-tumor reactivity of the infused TIL from NCT01814046 (Surgery Branch, NCI, Bethesda, MD, USA) was assessed by ELISA-based assays. Following overnight co-culture of the TIL with their autologous source tumor, the supernatant from these respective co-cultures was assessed by ELISA to determine the tumor-induced IFN-γ production as described previously[15].

### DNA extraction, library preparation, sequencing, and somatic analysis

Whole genome sequencing of the majority of samples was performed at the UPMC Genome Center ($n = 93$). Genomic DNA was isolated from tumor samples or peripheral blood mononuclear cells on the automated Chemagic 360 (PerkinElmer) instrument according to the manufacturer's instructions. Extracted DNA was quantitated using Qubit dsDNA BR Assay Kit (Thermo Fisher Scientific). DNA libraries were prepared using the KAPA Hyper Plus Kit (KAPA Biosystems). Genomic DNA was processed through fragmentation, enzymatic end-repair and A-tailing, ligation, and quality check a Standard Sensitivity NGS Fragment Analyzer Kit (Agilent). Libraries with an average size of 450 base pairs (range = 300–600 base pairs) were quantified by qPCR on the LightCycler 480 (Roche) using the KAPA qPCR quantification kit (KAPA Biosystems). The libraries were normalized and pooled as per manufacturer protocol (Illumina). Sequencing was performed using the NovaSeq 6000 platform (Illumina) with 151 base pair paired end reads to an average target depth of 70X coverage. The sequencing data was demultiplexed with bcl2fastq2 v2.20 (Illumina) to produce the fastq files.

The samples were mapped with Sentieon v1.3.4[62]. Somatic variants were called by TNhaplotyper2 on tumor-normal mode with the best-practice recommended whole genome sequencing setting. Variants were annotated with Funcotator from GATK v4.0.5[63]. Copy number alterations were called with an in-house developed ensemble method (CNVsenate) with mapped BAM and somatic SNV VCF files. CNVsenate gathers calling results from GATK v4.0.5[63], CNVkit v0.9.5[64], CNVnator v0.2.7[65], Manta v1.3.2[66], Sentieon CNV (201911)[62] and combines calling with SURVIVOR2 v1.0.3[67], then uses machine learning and event size for filtering. The filtered results were annotated with AnnotSV v1.1.1[68] for affected genes. The denoised copy ratio for chromosomal segments from GATK was primarily used. A customized script was used to calculate the denoised copy ratio for chromosomal arms. Copy number gain was defined as chromosomal arm denoised copy ratio ≥1.25, while copy number loss was defined as chromosomal arm denoised copy ratio ≤0.80.

Somatic SNV VCF files were converted to MAF format with vcf2maf v1.6.19[69] and annotated with VEP v102[70]. The MAF cohort was

filtered with a genomic data commons-like strategy, including for population allele frequency <2%, coding regions, and presence in dbSNP[71] and COSMIC[72]. Further filtering was done for only somatic mutations with variant allele frequency ≥5%. Mutations were manually tabulated for one sample that was unable to be processed into the MAF format (UM #20). All computational processes above were performed on a linux-based amazon web services ec2 instance on the DNAnexus platform (DNAnexus). The MAF file was then processed and summarized using maftools v2.10.05[73].

For six samples (UM #4, #13, #22, #23, #26 and #30) without sufficient tumor tissue for whole genome sequencing we utilized whole exome sequencing performed as described previously[14] to assess for canonical UM somatic mutations. For one sample (UM #53) without sufficient tumor tissue for whole genome sequencing DNA and RNA were extracted from paraffin embedded tumor tissue and processed with the Oncomine Comprehensive Assay v3 DNA and RNA primer sets (Thermo Fisher Scientific) according to the manufacturer's protocol. Alterations assessed were per the UPMC Oncomine panel which has been described previously[74].

### RNA extraction, library preparation, sequencing, and read alignment

Total RNA was isolated from tumor samples on the automated Chemagic 360 (PerkinElmer) instrument according to the manufacturer's instructions. Extracted RNA was quantitated with the Qubit RNA BR Assay Kit (Thermo Fisher Scientific) followed by an RNA quality check using Fragment Analyzer (Agilent). For each sample, RNA libraries were prepared from 100 ng of RNA using the KAPA RNA HyperPrep Kit with RiboErase (Kapa Biosystems) according to the manufacturer's protocol, followed by a quality check using Fragment Analyzer (Agilent) and quantification by qPCR with the Kapa qPCR quantification kit (Kapa Biosystems). The libraries were normalized, pooled, and sequenced using the NovaSeq 6000 platform (Illumina) to an average of ~50 million 101 base pair paired end reads. The sequencing data was demultiplexed with bcl2fastq2 v2.20 (Illumina) to produce the fastq files.

### Bulk transcriptomic computational analyses

Sequencing data was quality controlled with FastQC v0.11.7[75] before and after adapter trimming with cutadapt v1.18[76] along with assessment of estimated ribosomal content with sortmerna v4.3.4[77]. Trimmed reads were then aligned with STAR v2.7.5a[78] using the Gencode v38 GTF and GRCh38 fasta references[79]. Uniquely mapped percentage of reads and total uniquely mapped reads metrics after STAR mapping were used as further quality control metrics. The BAM file was indexed with samtools v1.10[80]. Gene counts from the STAR BAM files were calculated with htseq-count v0.13.5[81].

Gene names were converted from Ensembl v103[82] to HUGO gene symbols with biomaRt v2.50.3[83]. Redundant gene counts after name conversion were summed. Transcripts per million (TPM) were calculated in standard fashion using gene lengths calculated with FeatureCounts v1.6.2[84]. Raw counts were normalized with DESeq2 v1.34.0[85] using default and recommended parameters. Variance stabilizing transformation was performed on the normalized counts and used for principal component analysis (PCA) with PCAtools v2.10.0[86]. PCA was performed using the 10% most variant genes ($n = 5942$) in the dataset. Differential gene expression by UMIS level was performed without any adjustment parameters with default and recommended settings.

Enrichment scores of gene sets were calculated with singscore v1.14.0[18] using TPM input. Calculations utilized the unidirectional expected-upregulated mode, with the exception of the immune resistance program score[33] which was calculated using the bidirectional mode using separate expected-upregulated and expected-downregulated gene sets. UMIS was calculated with singscore using the unidirectional expected-upregulated mode using with the 2394

genes that positively correlated with immune and inflammatory hallmark gene set enrichment (negative PC2 gene loading) (Supplementary Data 4). A cohort-dependent version of UMIS was also calculated using gene set variation analysis (GSVA) with GSVA v1.42.0[87] using default settings and the same list of genes (Supplementary Data 4). This was only done for the purposes of comparison to the cohort-independent implementation with singscore and was not used elsewhere. Functional annotation of genes within UMIS ($n = 2394$) was performed with the Database for Annotation, Visualization and Integrated Discovery (DAVID) online tool (accessed March 20th, 2022)[88] after filtering for protein coding genes using Human Genome Organization (HUGO) Gene Nomenclature Committee (HGNC) complete set annotation (accessed October 6th, 2021). Similarly, functional annotation of differentially expressed genes between UMIS levels was performed with clusterProfiler v4.2.2[89] using the fgsea v3.16 method on only protein-coding genes using HGNC complete set annotation (accessed October 6th, 2021). Correlation and clustering analysis of PCs used the Human Molecular Signatures Database Hallmark gene set collection[90] while functional annotation used the Human Molecular Signatures Database Gene Ontology Biological Process gene set collection[91].

Human leukocyte antigen (HLA) typing of patients was performed using tumor bulk total RNAseq data. The arcasHLA v0.5.0[92] package was run with default settings to produce an output of genotypes for samples. Representative data for patients with multiple tumor samples was selected using the same algorithm as described previously in the survival analysis. TCR repertoires of tumors were analyzed from bulk total RNAseq data using MiXCR v3.0.12[93] with allowPartialAlignments=true as recommended for bulk RNAseq data. Counts were tabulated per amino acid CDR3 clonotype and used to calculate diversity (Shannon index) and clonality (1–Pielou's index) for TRB and TRA chains[42]. Samples with one or zero detected unique clonotypes were excluded from diversity and clonality analysis due to mathematically undefinable clonality; this resulted in exclusion of 12 metastases from TRB analysis and 18 metastases from TRA analysis. Public versus private repertoire analysis was performed using immunarch v0.6.9[94].

### Targeted TCR repertoire library preparation, sequencing, and analysis

Targeted TCR repertoires of paired tumors and post-REP TIL were derived from respective total RNA. Libraries were prepared using the QIAseq Immune Repertoire RNA Library Kit (Qiagen) per manufacturer's instructions. Libraries underwent quality check using a Standard Sensitivity NGS Fragment Analyzer Kit (Agilent) and quantification by qPCR with the Kapa qPCR quantification kit (Kapa Biosystems). The libraries were normalized, pooled, and sequenced using the MiSeq platform (Illumina) to an average of ~2.5 million 251 base pair paired end reads. The sequencing data was demultiplexed with bcl2fastq2 v2.20 (Illumina) to produce the fastq files. Sequencing data was processed using the Qiagen Biomedical Genomics Analysis 23.0 (Qiagen) per default recommended settings. Counts were tabulated from output files per amino acid CDR3 clonotype and used to calculate diversity (Shannon index) and clonality (1–Pielou's index) for TRB and TRA chains.

### Single cell RNA sequencing library preparation and sequencing

Selected tumors and previous treatments were UM #72 (tebentafusp), 83 (ICI and tebentafusp), 100 (ICI), 46 (liver directed therapy), 79 (cytotoxic chemotherapy and ICI), and 80 (liver directed therapy, kinase inhibition, antiangiogenic therapy and ICI). Cryopreserved single cell suspensions of selected tumors were prepared for input into the Chromium Next GEM Single Cell 5' Reagent Kit v2 (10X Genomics) by thawing in complete media with human AB serum, sequential filtration through 70 mm and 30 mm MACS SmartStrainers (Miltenyi Biotec) and removal of dead cells using the Dead Cell Removal Kit

(Miltenyi Biotec) per manufacturer's protocol. Cell suspensions were inspected to confirm adequate viability (≥70%). Each tumor sample was processed in a separate lane of the Chip K, with ~35,000 cells loaded per sample. The 5' gene expression and TCR V(D)J libraries were then prepared per manufacturer's instructions. Prior to sequencing, libraries underwent quality check using a Standard Sensitivity NGS Fragment Analyzer Kit (Agilent) and quantification by qPCR with the Kapa qPCR quantification kit (Kapa Biosystems). The libraries were normalized, pooled, and sequenced using the NovaSeq6000 platform (Illumina) to an average of ~30,000 paired-end reads per 5' gene expression library per cell and ~5000 paired-end reads per TCR V(D)J library per cell with parameters per manufacturer's protocol. The sequencing data was demultiplexed with bcl2fastq2 v2.20 (Illumina) to produce the fastq files.

## Single cell RNA sequencing computational processing

Sequencing data was processed using 10X Genomics Cell Ranger multi v6.1.2[95] using 10X Genomics Cloud Analysis with introns excluded and an estimated expected cell count of 20,000. Bioinformatic processing of each sample involved adjustment for ambient RNA contamination with SoupX v1.5.2[96] (default settings), normalization with the sctransform v2 method within Seurat 4.1.1[97,98] (default settings), and estimation and removal of doublets with DoubletFinder v2.0.3[99] (default settings). Cells remaining after quality control and removal of doublets were then input into our cataloging algorithm. This involved first assigning cells to large buckets using UCell v2.1.0[20] in the following order: immune (UCell score >0 for gene set of *PTPRC*), tumor (UCell score >0 for gene set of *SOX10, S100A1, MITF, MLANA, PMEL, TYR*), stroma (all remaining cells). A TIL atlas was created using a published dataset[19] with harmony v0.1.0[100] and symphony v0.1.0[21] using settings appropriate for the normalization method of the published dataset. Our samples' immune fractions were then mapped onto the atlas using symphony settings recommended for data normalized with sctransform v2. The higher resolution "level 2" annotation, which included 31 phenotypes, was utilized. The addition of tumor and stroma cells to these immune cells completed our cellular cataloging and various levels (overall, broad, granular) were also assigned to the cells (Supplementary Data 9). For pooled analysis, samples were integrated with Seurat 4.1.1 using settings appropriate for sctransform v2-normalized data. Dimensionality reduction and differential gene expression were performed on the integrated Seurat object. Counts of specific cell types were derived from this integrated Seurat object. Comparison of cell type proportions by UMIS level was performed with the propeller function within the speckle v0.0.3[101] package using the arcsin square root transformation of proportions method.

TCR V(D)J repertoires were filtered for most frequent TRA and TRB chains using scRepertoire v1.4.0[102]. Counts were tabulated for cells with paired TRA and TRB chains per unique TRA-TRB amino acid CDR3 clonotype and used to calculate diversity (Shannon index) and clonality (1−Pielou's index).

## Statistical analysis

Statistics were calculated using R v4.1.2[103] (R Core Team) with RStudio v.2022.12.0 + 353[104] (Rstudio Team) or GraphPad Prism v9.5.0 (GraphPad Software), and specific statistical analyses used are highlighted in the respective figure legends. In general, continuous-continuous associations were assessed with the Spearman's rank correlation with simple linear regression with 95% confidence intervals only to illustrate linearity. Unpaired categorical-continuous associations were assessed with the Wilcoxon rank-sum test (two-tailed) or Kruskal–Wallis one-way analysis of variance test as appropriate. Paired categorical-continuous associations were assessed with the Wilcoxon signed-rank test (two-tailed). Categorical-categorical associations were assessed with the Fisher's exact test. Receiver operating characteristic curves were generated using

univariate logistic regression and mapping of true positive 1 − specificity versus sensitivity. Areas under the ROC curves were calculated using the trapezoid rule. Time-to-event curves using the Kaplan–Meier method were generated with survminer v0.4.9[105] and comparisons between categorical groups were assessed with the logrank test. Clustering analysis was performed with Complex-Heatmap v2.10.0[106] and used the default method of Euclidean distance. Where appropriate, multiple comparison adjustment was performed with the false discovery rate (FDR) method using the *p.adjust* function with method = "fdr" in R. In R the lowest possible numeric value is roughly 1e−324. Thus, we presented values less than 1e−324 as ~0 rather than listing arbitrary lower limit numbers.

## Utility visualization software

Aside from software previously mentioned, the following were used for various visualizations throughout the manuscript: tidyverse v1.3.2[107], ggplot2 v3.4[108], RColorBrewer v1.1-3[109], ggprism v1.0.4[110], patchwork v1.1.2[111], packcircles v0.3.4[112], plotly v4.10.0.9001. Illustrations were created with BioRender.com.

## Reporting summary

Further information on research design is available in the Nature Portfolio Reporting Summary linked to this article.

## Data availability

Bulk total RNAseq raw sequencing data generated in this study have been deposited in the database of Genotypes and Phenotypes (dbGaP) under accession number phs003330.v1.p1 [https://www.ncbi.nlm.nih.gov/projects/gap/cgi-bin/study.cgi?study_id=phs003330.v1.p1]. These data are available under restricted access for patient confidentiality reasons and access can be obtained by request via the dbGaP system by following the instructions provided by the website. Approval is determined by the National Cancer Institute Data Access Committee, which can be emailed at ncidac@mail.nih.gov. Access to data is generally granted within a month of successful application and available indefinitely thereafter. Selected raw data are protected and are not publicly available due to data privacy laws but may be shared upon request. Source data are provided with this paper.

## Code availability

Software packages were implemented as described in the "Methods" and no custom packages were created.

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

## Acknowledgements

We thank the members of the UPMC clinical trial and research teams for their efforts in this study and the Surgery Branch, NCI, for providing selective tumor samples. We thank Clinigen for providing interleukin-2 for clinical and research studies and thank all the patients who participated in this study. This research was supported by the UPMC Immune Transplant and Therapy Center and in part by the University of Pittsburgh Center for Research Computing, RRID:SCR_022735, through the resources provided. Specifically, this work used the HTC cluster, which is supported by NIH award number S10OD028483. This work utilized the UPMC Hillman Cancer Center Immunologic Monitoring and Cellular Products Laboratory, a shared resource at the University of Pittsburgh supported by the CCSG P30 CA047904. S.L.-M. was supported by a National Institutes of Health National Cancer Institute training grant (T32CA113263).

## Author contributions

S.L.-M., C.B., G.Y., S.M., C.G., J.T., E.A. and P.M. carried out the experiments. S.L.-M., R.K. and U.K. analyzed data and prepared visualizations. S.L.-M., P.M., A.C., V.S., P.C. and U.C. designed the RNAseq bioinformatic pipeline. S.L.-M., P.M., E.S., G.V., L.K., D.K. and X.Z. performed next-generation sequencing. U.K. conducted associated TIL ACT clinical trials. Y.-M.H. and U.K. supervised TIL manufacturing associated with TIL ACT clinical trials. S.L.-M. collected clinicogenomic patient and metastasis data. S.L.-M. and U.K. conceived the project, designed the experiments, and wrote the manuscript.

## Competing interests

The authors declare no competing interests.

## Ethical approval

We have complied with all relevant ethical regulations in the conduct and reporting of this study.
