## [Peer Review File · Nature Communications]

Uveal Melanoma Immunogenomics Predict Immunotherapy Resistance and SusceptibilityReviewer #1 (Remarks to the Author): expert in TCR sequencing and immunology

In this study, Leonard-Murali S et al utilized immunogenomics approaches to identify gene signatures associated with immunotherapy responses against uveal melanoma. Overall, this study provided a valuable resource for this rare cancer type, suitable for this journal. Additionally, the UMIS prediction may present a significant value for patients with uveal melanoma. The experimental design and data interpretation are rigorous; therefore, I only have a few minor comments.

Minor comments:

1. For the single-cell analysis, please briefly describe the prior treatments for those 6 tumor specimens (UM# 72, 83, 100, 46, 80, 79). This information might have some impact for the data interpretation. Based on Figure 1c, it seems to me that patients #72 and #83 were treated with tebentafusp previously, while most patients were also treated with checkpoint inhibitors.
2. I have another comment related to the single-cell analysis. It's understandable that the low UMIS group had low numbers of infiltrating immune cells. Repeating single-cell analysis is likely not feasible for these precious samples. In the future, it will be great to enrich these cells by FACS first (such as sorting CD45+ cells). Therefore, the immune cell numbers between UMIS high and low will be more equal. It's better for the comparison in the single-cell analysis.
3. Please re-calculate the $\log_2(\text{fold change})$ numbers to "Fold Changes". It will be easier for the readers to appreciate the differences between groups.

Reviewer #3 (Remarks to the Author): expertise in uveal melanoma multi-omics

The work states that Immune Checkpoint Inhibition (ICI) therapy has had limited efficacy against most other solid tumors, especially those with low tumor mutational burden (TMB). Uveal melanoma (UM), a prototypic ICI resistant cancer with low TMB, is a focused case study of the authors. The goal of the work is to develop more effective immunotherapeutics for metastatic UM based on their previous observations (i.e., a group of UM metastases naturally harbor TIL with potent autologous anti-tumor reactivity and ACT administering such TIL could efficiently mediate tumor regression). The authors conducted a thorough analysis of the immune system and genes in the largest and most varied collection of human UM metastases ever assembled.

Major

- Section "Clinicogenomic landscape of metastatic uveal melanoma". Statistical issues
 - + They said "No associations were found between TMB and cohort demographics" I consider they state this via the FDR because some associations are significant statistically with P-values < 0.05 (Here I set the alpha to 0.05). So, the authors must state thoroughly which method they use to compute FDR? How to compute this?
 - + I wonder whether it is right to compute the FDR and/or PCA or not for some reasons:
 1. In the case of a study with only 100 metastases, meaning the number of hypotheses being tested may be relatively small.
 2. Moreover, the sentence "One hundred metastases were surgically procured from 84 UM patients" makes 100 metastases can be considered dependent to some extent, leading to a serious problem that is the authors should not use the FDR and PCA.
- Organization of the manuscript is very poor. The authors should remember that the Results section is above the Methods one, so they need to organize the contents properly. Besides, the authors usually put all applied methods into the title or legend of Figures instead of thoroughly stating in the content of the Result section. These makes the readers to be hard to follow and understand due to lack of important information. For example:
 - + "To determine whether specific cellular pathways and processes were associated with specific PCs, we correlated PC coordinates (1, 2, and 3) with enrichment scores for each of the canonical hallmark gene sets from the Human Molecular Signatures Database (Supplementary Table 2)". Which tool did the authors use to render the results sitting in Supplementary Table 2? Please explain the results and give some comments (if needed) in the manuscript instead of showing the

table in Supplementary Table 2 only. I guess Supplementary Table 2 showed the values of each PC coordinate along 100 metastases in the three first columns (the "UM #" column does not count), and the rest of the columns are the Spearman's rank correlation coefficients (ρ) between three PCs and hallmark gene set collection enrichment scores, didn't it? They should describe what exactly did they do to make the work reproducible.

+ "Unsupervised clustering of the PC-gene set correlations identified 4 discrete clusters (A, B, C, and D) with unique biologic motifs (Fig. 2a)". Very ambiguous. What exactly clustering method / distance did you use to cluster PC-gene set correlations (figure 2a)? I guess the author first correlated PC coordinates with enrichment of hallmark signatures for each individual metastatic sample ($n=100$) using Spearman's rank correlation coefficients (ρ). They then hierarchically clustered those rhoes with the Euclidean distance.

+ The individual metastatic samples were further clustered by their relative expression of each of the hallmark gene set clusters to reveal striking variability across the tumor cohort (Fig. 2b). Were the z-scores shown in the heatmap normalized from "their relative expression of each of the hallmark gene set clusters"? what does "their relative expression of each of the hallmark gene set clusters" mean here? Expression levels of genes belonging to each of the 4 clusters? How many genes in total? How many did genes sit in each cluster?

+ "First, we defined the 2394 genes that positively correlated with immune and inflammatory hallmark gene set enrichment (negative PC2 loading)." Again, the author identified the 2394 genes in PC2 that positively correlated with immune and inflammatory hallmark gene set enrichment. How did they do to discover these genes? If the authors described how to do in the Methods section, they should cite that here.

- If needed, the authors can run a survival analysis on the four clusters (A, B, C, D). The survival analysis would likely show that group B had a significantly worse outcome than the other three clusters. This possibly strengthens the authors' focus on immune systems for this case study.

Reviewer #4 (Remarks to the Author): expert in uveal melanoma immunogenomics

This study provides new and important insights of resistance to immuncheckpoint blockade and the T-cell engager tebentafusp in patients with metastatic uveal melanoma. The authors have done an extensive analysis of the immunogenetic landscape, in what seems, a relevant and typical patient material that address both biologically and clinically relevant perspectives. Of special clinical importance is the immunogenetic profiling that suggests a method, UMIS, that could predict clinical response to autologous TIL therapy.

The genetic and immunological data are investigated using appropriate methods allowing for a detailed understanding. However, the presentation of clinical information can be improved, Fig 1C is not enough. There is no table of patient characteristics. I suggest including a conventional table with data for sex, age, performance status, subtype/localization of primary tumor, HLA-subtype, time to metastatic disease, time to medical treatment, primary site for metastases, LDH level, M-stage at diagnosis of metastases, M-stage and treatment level at biopsy, lines of treatment including other than medical, i.e., locoregional therapy.

In the material from 84 patients, in 100 tissues biopsies from different localizations, its stated that TILs could be identified in more than half. For the 50% where TILs could not be identified it would be also important to understand more. A more detailed information in suppl information on their characteristics, clinically and for the immunogenetic profile would helpful for future research and also for development of TIL therapy.

The correlation of UMIS level and clinical outcomes, Figure 8, is somewhat surprising and disappointing, is it possible to add eg. sequence of therapy in this data, and also introduce the variable "time to next treatment"?

Perhaps the most clinical interesting finding for TIL therapy development in terms of treatment

opportunities with TILs in a refractory patient, eg. Fig 10 and 11. A greater understanding of tumor reactive TILs is of special interest to learn how to enhance the activity of TIL products. The observations are interesting but builds on few numbers, N=2 for tebe and N=2 for both ICIs and tebe. It would be preferable to see more observations, so you get a reasonable number to understand the relevance, i.e. as it is for ICIs; N=5. Can you provide additional data from new patients? After all this data would be major interest in understanding sequencing TIL therapy, i.e. whether the profiling support clinical decision making or not.

Although there is a long list of references there are a number of relevant recent key references in this area of research that for unknown reasons are not cited. Please reconsider some as below.

Field et al. *Nat Commun.* 2018 Jan 9;9(1):116.

Shain et al. *Nat Genet.* 2019 Jul;51(7):1123-1130.

Karlsson et al. *Nat Commun.* 2020 Apr 20;11(1):1894.

Durante et al. *Nat Commun.* 2020 Jan 24;11(1):496.

Pelster et al. *J Clin Oncol.* 2021 Feb 20;39(6):599-607.

Lutzky et al., Phase 2 Trial of Nivolumab plus Relatlimab in Metastatic Uveal Melanoma. (*CA* 224-294). SMR 2022.

Mariani et al. *Br J Cancer.* 2023 Sep;129(5):772-781.

Carvajal et al. *Nat Rev Clin Oncol.* 2023 Feb;20(2):99-115.

Reviewer #5 (Remarks to the Author): expertise in immunology analysis

Despite the unprecedented therapeutic benefits of immunotherapies (esp. immune checkpoint blockade (ICB)) in patients with cutaneous melanoma (CM), patients with uveal melanoma (UM) have not been responding well to these novel immunotherapeutics, which has been attributed at least partially to UM being "cold". In this manuscript, Leonard-Murali, et al. reported that UMs are actually not cold, at least in UM patients screened for adoptive cell therapy (ACT) with tumor-infiltrating T cells (TILs), as more than 50% of the metastases harvested in these patients have tumor-reactive TILs. Instead, the team reason that the lack of response in UM patients to ICB and tebentafusp (a bispecific glycoprotein 100 peptide-HLA-directed CD3 T cell engager) is due to the extrinsic factors in UM that suppress the expansion of these reactive TILs. In support of this, the investigators showed that TILs isolated from UMs can be successfully expanded ex vivo, which, upon reinfusion back to the patients, can lead to appreciable tumor reduction. To predict which patients/metastases contain these TILs, the authors developed an immune metric called uveal melanoma immunogenomic score (UMIS), based on 2394 genes (1527 protein-coding genes and 867 non-coding/unclassified/pseudo genes) derived from Principal Component 2 (PC2).

Subsequent evaluations of UMIS showed that metastases with UMIS <0.2 rarely produced reactive TIL cultures, indicating 0.2 is a preoperative threshold for surgical resection of "good and productive" metastases as sources of TIL preparation. Furthermore, UMIS above 0.246 predicted a significantly improved progression-free and overall survival of patients receiving adoptively transferred TILs. While these results support a threshold UMIS as a criterion in selecting metastases from UM patients for TIL expansion, some concerns remain:

1. While UMIS may be useful in identifying metastases for better TIL expansion, it is noteworthy to mention that most of times it is not about which metastases to resect but rather about whether all the resectable metastases would provide enough materials for TIL expansion. Moreover, although UMIS inversely correlated with a reported immune resistance program (Fig. 4j), it did not predict/correlate ICB responses, nor did it predict a better overall survival.
2. While it is plausible to reason that extrinsic factors in UM suppress TIL expansion in vivo, these factors remain to be defined, a description of which would strengthen this study.
3. Although UMIS was the strongest performer as both a correlative and classification metric for predicting ex vivo TIL reactivity, it remains to be determined whether it is really "significantly better" than other previously defined predictive biomarkers such as T cell GEP, Teff/IFN-g GEP, etc. This is important, given the complex composition of the 2394 genes used to calculate UMIS that would make it difficult to adopt in the clinic, particularly in other institutions.
4. Single random biopsies (i.e., central core fragments) from 100 metastases were utilized in this study to "facilitate a clinically relevant and minimally invasive biopsy approach for in situ tumor". A potential concern with this approach is how representative these single biopsies are to whole

tumors. Additionally, further elaboration is needed on how biopsy of central core fragments would be done in situ.

5. Detailed description of how principal components (particularly, PC2) were defined should be provided, as this is key to the calculation of UMIS. Similarly, every cluster (A, B, C, or D) has all of the 100 studied metastases. How were these clusters (A-D) "clustered"?

6. Minor points: Line 29, "began focused studies" may read better with "focused our studies"; please clarify "~0" on Fig. 4i; please elaborate more on how autologous APCs were prepared (Fig. 5b).

Uveal Melanoma Immunogenomics Predict Immunotherapy Resistance and Susceptibility

Reviewer #1 (Remarks to the Author): expert in TCR sequencing and immunology

In this study, Leonard-Murali S et al utilized immunogenomics approaches to identify gene signatures associated with immunotherapy responses against uveal melanoma. Overall, this study provided a valuable resource for this rare cancer type, suitable for this journal. Additionally, the UMIS prediction may present a significant value for patients with uveal melanoma. The experimental design and data interpretation are rigorous; therefore, I only have a few minor comments.

Minor comments:

1. For the single-cell analysis, please briefly describe the prior treatments for those 6 tumor specimens (UM# 72, 83, 100, 46, 80, 79). This information might have some impact for the data interpretation. Based on Figure 1c, it seems to me that patients #72 and #83 were treated with tebentafusp previously, while most patients were also treated with checkpoint inhibitors.

Thank you for clarifying this observation. Yes, UM#72 and UM#83 both received prior tebentafusp and may have contributed to greater T cell infiltration into these metastases, as we describe in the text (lines 354-357) and demonstrate with TCR diversity analysis in Fig. 6c (see below) and Extended Data Fig. 10d.

“In contrast, prior tebentafusp treatment (n=12) was associated with greater TCR diversity, consistent with the ability of this bispecific T cell engager to recruit T cells to these metastases (Fig. 6c; Extended Data Fig. 10a,d,e).”

All but one patient had received prior checkpoint inhibition. To clarify the prior treatments, we have added the below sentence to the methods (lines 900-902).

“Selected tumors and previous treatments were UM #72 (tebentafusp), 83 (ICI and tebentafusp), 100 (ICI), 46 (liver directed therapy), 79 (cytotoxic chemotherapy and ICI), and 80 (liver directed therapy, kinase inhibition, antiangiogenic therapy and ICI).”

2. I have another comment related to the single-cell analysis. It's understandable that the low UMIS group had low numbers of infiltrating immune cells. Repeating single-cell analysis is likely not feasible for these precious samples. In the future, it will be great to enrich these cells by FACS first (such as sorting CD45+ cells). Therefore, the immune cell numbers between UMIS high and low will be more equal. It's better for the comparison in the single-cell analysis.

Thank you for this suggestion. We agree that CD45+ enrichment could be a valuable strategy to employ in the future as an enrichment step to allow for more detailed differential expression analyses in specific cell types. However, in this landscape report, we wanted to convey both the frequency of individual cell types and their concomitant transcriptomic expression. Further, we wanted to avoid the introduction of selection bias using purification approaches (i.e. CD45+) which may enrich for “healthier” populations and induce artifactual transcriptomic changes with the added manipulation. We have added language to clarify our approach (lines 178-182).

“Our single cell analysis of low UMIS tumors had expectedly low numbers of immune cells. However, to maintain the true proportional landscape of specific cell types and avoid manipulation induced transcriptomic changes, we profiled the tumor digests without an additional enrichment step.”

3. Please re-calculate the log₂(fold change) numbers to “Fold Changes”. It will be easier for the readers to appreciate the differences between groups.

Thank you for this suggestion. To aid the reader, we have added both “fold change”, in addition to “log₂(fold change)” to the results text describing the differences in *CTNNB1* and *SNHG7* expression (lines 229-231; lines 236-240) as follows:

“In contrast, low UMIS tumor cells had 1.44-fold higher expression of CTNNB1 (log₂(fold change)=-0.53, FDR ~ 0) which encodes the beta-catenin protein (Fig. 4h,i; Supplementary Table 12).”

“Interestingly, the most upregulated gene in low UMIS tumor cells was the long non-coding RNA, SNHG7, which was 3.48-fold upregulated in low UMIS tumor

cells ($\log_2(\text{fold change})=-1.80$, $FDR \sim 0$) and has been previously reported as a positive regulator of CTNNB1 expression in several cancers (Fig. 4h,i; Supplementary Table 12)³⁴⁻³⁸.

However, since $\log_2(\text{fold change})$ are the units employed for our violin plots, we maintained this unit nomenclature in the figures to avoid reader confusion.

Reviewer #3 (Remarks to the Author): expertise in uveal melanoma multi-omics

The work states that Immune Checkpoint Inhibition (ICI) therapy has had limited efficacy against most other solid tumors, especially those with low tumor mutational burden (TMB). Uveal melanoma (UM), a prototypic ICI resistant cancer with low TMB, is a focused case study of the authors. The goal of the work is to develop more effective immunotherapeutics for metastatic UM based on their previous observations (i.e., a group of UM metastases naturally harbor TIL with potent autologous anti-tumor reactivity and ACT administering such TIL could efficiently mediate tumor regression). The authors conducted a thorough analysis of the immune system and genes in the largest and most varied collection of human UM metastases ever assembled.

Major

- Section "Clinicogenomic landscape of metastatic uveal melanoma". Statistical issues + They said "No associations were found between TMB and cohort demographics" I consider they state this via the FDR because some associations are significant statistically with P-values < 0.05 (Here I set the alpha to 0.05). So, the authors must state thoroughly which method they use to compute FDR? How to compute this?

We utilized the "false discovery rate" method first proposed by Benjamini and Hochberg in 1995 (*Controlling the false discovery rate: a practical and powerful approach to multiple testing*. J. Roy. Statist. Soc. Ser. B57(1995), no.1, 289–300.) for all descriptions of FDR. The method to compute this is intricately described in the aforementioned paper, and the method is also available in most statistical programs and packages. We used the "p.adjust" function from base R, using method = "fdr", which is equivalent to method = "BH", or Benjamini-Hochberg. We have added clarifying language to the methods section to reflect this (lines 966-968).

"Where appropriate, multiple comparison adjustment was performed with the false discovery rate (FDR) method using the p.adjust function with method = "fdr" in R."

+ I wonder whether it is right to compute the FDR and/or PCA or not for some reasons:
1. In the case of a study with only 100 metastases, meaning the number of hypotheses being tested may be relatively small.

We used FDRs wherever applicable in the interest of adding stringency to our exploratory findings. PCA was done initially as an exploration but yielded such valuable

insights that we continued with our downstream analyses based upon the PCA observations.

2. Moreover, the sentence "One hundred metastases were surgically procured from 84 UM patients" makes 100 metastases can be considered dependent to some extent, leading to a serious problem that is the authors should not use the FDR and PCA.

Since our analyses focused on tumor characteristics, rather than patient-specific variables, our statistical experts felt that FDR and PCA were valid. However, to address this concern, we have added clarifying language regarding potential limitations (lines 504-506).

"A subset of patients contributed multiple metastases to the study (14 patients with multiple metastases included). We chose to include these metastases to maximize sample size in this rare cancer but recognize that this may be a source of selection bias."

- Organization of the manuscript is very poor. The authors should remember that the Results section is above the Methods one, so they need to organize the contents properly. Besides, the authors usually put all applied methods into the title or legend of Figures instead of thoroughly stating in the content of the Result section. These makes the readers to be hard to follow and understand due to lack of important information. For example:

+ "To determine whether specific cellular pathways and processes were associated with specific PCs, we correlated PC coordinates (1, 2, and 3) with enrichment scores for each of the canonical hallmark gene sets from the Human Molecular Signatures Database (Supplementary Table 2)". Which tool did the authors use to render the results sitting in Supplementary Table 2? Please explain the results and give some comments (if needed) in the manuscript instead of showing the table in Supplementary Table 2 only. I guess Supplementary Table 2 showed the values of each PC coordinate along 100 metastases in the three first columns (the "UM #" column does not count), and the rest of the columns are the Spearman's rank correlation coefficients (ρ) between three PCs and hallmark gene set collection enrichment scores, didn't it? They should describe what exactly did they do to make the work reproducible.

Thank you for these comments. The first 3 columns after UM # are each samples' PC coordinate value. However, the remainder of Supplementary Table 2 present enrichment scores of each UM for each of the listed gene sets from the hallmark gene set collection. The singscore package was used to make these calculations, which are derived from TPM, cohort independent, and are not related to the PCA coordinates. Correlating the PC coordinate columns with each of the gene set enrichment score columns yields the Spearman's ρ values that populate the cells within the heatmap for clustering (Fig. 2a). PCAtools was used to generate the PC coordinates and singscore was used for the enrichment scores. These are stated in the methods (lines 841-854).

“Variance stabilizing transformation was performed on the normalized counts and used for principal component analysis (PCA) with PCAtools v2.10.0⁸⁶. PCA was performed using the 10% most variant genes (n=5942) in the dataset. Differential gene expression by UMIS level was performed without any adjustment parameters with default and recommended settings.

Enrichment scores of gene sets were calculated with singscore v1.14.0¹⁸ using TPM input. Calculations utilized the unidirectional expected-upregulated mode, with the exception of the immune resistance program score³³ which was calculated using the bidirectional mode using separate expected-upregulated and expected-downregulated gene sets. UMIS was calculated with singscore using the unidirectional expected-upregulated mode using with the 2394 genes that positively correlated with immune and inflammatory hallmark gene set enrichment (negative PC2 gene loading) (Supplementary Table 4).”

+ “Unsupervised clustering of the PC-gene set correlations identified 4 discrete clusters (A, B, C, and D) with unique biologic motifs (Fig. 2a)”. Very ambiguous. What exactly clustering method / distance did you use to cluster PC-gene set correlations (figure 2a)? I guess the author first correlated PC coordinates with enrichment of hallmark signatures for each individual metastatic sample (n=100) using Spearman’s rank correlation coefficients (ρ). They then hierarchically clustered those rhoes with the Euclidean distance.

Thank you for these comments. Yes, your description is accurate. To provide the reader clarity, this approach has been described in the methods and clarified in the results (lines 96-99).

“To determine whether specific cellular pathways and processes were associated with specific PCs, we correlated PC coordinates (1, 2, and 3) with enrichment scores for each of the canonical hallmark gene sets from the Human Molecular Signatures Database (Supplementary Table 3)¹⁸. Unsupervised clustering (Euclidean distance) of the PC-gene set correlations (Spearman’s ρ) identified 4 discrete clusters (A, B, C, and D) with unique biologic motifs (Fig. 2a).”

“Clustering analysis was performed with ComplexHeatmap v2.10.0¹⁰⁶ and used the default method of Euclidean distance.”

+ The individual metastatic samples were further clustered by their relative expression of each of the hallmark gene set clusters to reveal striking variability across the tumor cohort (Fig. 2b). Were the z-scores shown in the heatmap normalized from “their relative expression of each of the hallmark gene set clusters”? what does “their relative expression of each of the hallmark gene set clusters” mean here? Expression levels of genes belonging to each of the 4 clusters? How many genes in total? How many did genes sit in each cluster?

Thank you for these comments. We used singscore absolute enrichment scores, then calculated Z-scores within each pathway. The Z-scores themselves are the “relative”

metric. The genes and number within each pathway are open source and available on the MSigDB website (<https://www.gsea-msigdb.org/gsea/msigdb/human/genesets.jsp?collection=H>), which is referenced. The pathways are associated with each cluster, rather than individual genes.

+ “First, we defined the 2394 genes that positively correlated with immune and inflammatory hallmark gene set enrichment (negative PC2 loading).” Again, the author identified the 2394 genes in PC2 that positively correlated with immune and inflammatory hallmark gene set enrichment. How did they do to discover these genes? IF the authors described how to do in the Methods section, they should cite that here.

Thank you for these comments. We used PCA loadings to determine which genes correlated with increased enrichment of immune and inflammatory pathways (negative PC2 loadings). This is the specific purpose of Fig. 2c, which found the unique relationship between cluster B and PC2. We have added clarifying language to the results section “*Development of Uveal Melanoma Immunogenomic Score (UMIS)*” where it previously mentioned negative PC2 loading. Relevant text is shown below (lines 111-121).

“Having identified transcriptomic differences among the metastases, we next sought to determine whether any of the three PCs independently correlated with the expression of the gene set clusters. Average Spearman’s rank correlation coefficients (ρ) for each of the gene set cluster enrichment scores (A, B, C, and D) were mapped against the individual PCs (1, 2, and 3) (Fig. 2c). We observed that cluster A (cellular metabolism) was strongly correlated with PC3 (mean $\rho=+0.76$) but also weakly correlated with the negative aspect of PC1 (mean $\rho=-0.27$). Cluster B (immune and inflammatory signaling) was exclusively correlated with the negative aspect of PC2 ($\rho=-0.32$). Clusters C and D were not found to independently correlate with any of the three PCs.”

- If needed, the authors can run a survival analysis on the four clusters (A, B, C, D). The survival analysis would likely show that group B had a significantly worse outcome than the other three clusters. This possibly strengthens the authors' focus on immune systems for this case study.

Thank you for these comments. The clusters are not clusters of tumors or patients, but rather clusters of pathways. Therefore, survival analysis of cluster versus cluster is not feasible. However, to address this concern we include survival analysis (Extended Data Fig. 9a) of patients using UMIS as a categorical variable.

Reviewer #4 (Remarks to the Author): expert in uveal melanoma immunogenomics

This study provides new and important insights of resistance to immuncheckpoint blockade and the T-cell engager tebentafusp in patients with metastatic uveal melanoma. The authors have done an extensive analysis of the immunogenetic landscape, in what seems, a relevant and typical patient material that address both

biologically and clinically relevant perspectives. Of special clinical importance is the immunogenetic profiling that suggests a method, UMIS, that could predict clinical response to autologous TIL therapy.

The genetic and immunological data are investigated using appropriate methods allowing for a detailed understanding. However, the presentation of clinical information can be improved, Fig 1C is not enough. There is no table of patient characteristics. I suggest including a conventional table with data for sex, age, performance status, subtype/localization of primary tumor, HLA-subtype, time to metastatic disease, time to medical treatment, primary site for metastases, LDH level, M-stage at diagnosis of metastases, M-stage and treatment level at biopsy, lines of treatment including other than medical, i.e., locoregional therapy.

Thank you for this suggestion. We have added a comprehensive table of patient characteristics (Supplementary Table 1) that summarizes the clinicogenomic variables shown in Fig. 1c including sex, age at diagnosis, AJCC M stage, metastatic tumor extent, ocular tumor treatment, alkaline phosphatase level, lactate dehydrogenase level, prior therapies (including regionally with liver directed therapies), metastatic site, and canonical genomic mutation and copy number alterations. Performance status of all patients was ECOG 0 or 1 as determined by eligibility screening for TIL therapy.

In the material from 84 patients, in 100 tissues biopsies from different localizations, its stated that TILs could be identified in more than half. For the 50% where TILs could not be identified it would be also important to understand more. A more detailed information in suppl information on their characteristics, clinically and for the immunogenetic profile would helpful for future research and also for development of TIL therapy.

Thank you for these comments. We completely agree that this poorly immunogenic subset of metastases is extremely important in defining underlying factors that promote and inhibit immunogenicity in these metastases. Tables 5 and 13 present all the clinicogenomic variables as they relate to UMIS level and TIL reactivity, demonstrating no associations between these variables and the immune metrics (all $P > 0.05$) This information is included as follows (lines 157-161; lines 282-285):

“UMIS level was observed to be independent of metastatic site (Fig. 3g, Supplementary Table 5), TMB (Fig. 3h, Supplementary Table 4), somatic mutations and copy number alterations (Extended Data Fig. 3b, Supplementary Table 6,7), and class I human leukocyte antigen (HLA) alleles (Extended Data Fig. 3c).”

“The percentage of tumor reactive TIL cultures was also independent of metastatic site, TMB, specific mutation expression, copy number alterations, and class I HLA alleles (Extended Data Fig. 7a,b,c,d; Supplementary Table 13,14,15).”

Further, single cell RNAseq analysis was performed with the goal of uncovering unique immune evasion mechanisms in these poorly immunogenic tumors. As demonstrated in Fig. 4i, we identified *SNHG7* and beta catenin as potential drivers of immune exclusion (see below)

The correlation of UMIS level and clinical outcomes, Figure 8, is somewhat surprising and disappointing, is it possible to add eg. sequence of therapy in this data, and also introduce the variable “time to next treatment”?

Thank you for this comment. We were also surprised to see a lack of survival difference between patients with high versus low UMIS tumors (Extended Data Fig. 9a). However, we believe that this finding suggests UMIS reflects intrinsic immune responses that are actively suppressed within the tumor microenvironment. Indeed, based upon our analysis of survival of patients with variable UMIS tumors after TIL therapy, we found that high UMIS tumor patients had improved progression-free [$P = 0.044$] and overall survival [$P = 0.009$], Fig. 6j, shown below)

j
We did assess the sequence of prior checkpoint inhibition (single agent versus sequential versus combination) in the context of several dependent variables (TMB, UMIS, TIL reactivity, TCR diversity and TCR clonality; Fig. 7c, Extended Data Fig. 10d, Supplementary Tables 2, 5, 13). We did not find any specific therapeutic association with any of the above dependent variables. Time to next treatment could not be evaluated for the current cohort since accurate dates for prior treatments could not be confirmed for all patients.

Perhaps the most clinically interesting finding for TIL therapy development in terms of treatment opportunities with TILs in a refractory patient, eg. Fig 10 and 11. A greater understanding of tumor reactive TILs is of special interest to learn how to enhance the activity of TIL products. The observations are interesting but build on few numbers, N=2 for tebe and N=2 for both ICIs and tebe. It would be preferable to see more observations, so you get a reasonable number to understand the relevance, i.e. as it is for ICIs; N=5. Can you provide additional data from new patients? After all this data would be of major interest in understanding sequencing TIL therapy, i.e. whether the profiling supports clinical decision making or not.

Thank you for this suggestion. We have added data for an additional 10 patient tumors. This expanded cohort (n=19) includes 10 ICI treated tumors, 4 tebentafusp treated tumors, 2 tumors treated with both agents, and 3 which received neither. In this expanded cohort, we continue to observe increased clonality after *ex vivo* expansion across all of these samples, regardless of prior therapy (revised Fig. 6d, Extended Data Fig. 11a,b,c). These findings corroborate that the endogenous TILs were not limited by intrinsic proliferative deficiencies, but instead their growth was likely suppressed by the tumor microenvironment. Taken together, we observed the quiescence of endogenous TIL in UM metastases was not reversed with ICI or tebentafusp but could be revived with *ex vivo* liberation and expansion. In terms of clinical decision making, our study was not designed to inform sequence of therapy, but rather to better understand how

TIL therapy could induce cancer regression in UM patients when ICI and tebentafusp could not. The revised Fig. 6d is shown below:

d
Ex vivo TIL expansion from treatment naïve and refractory patients

Although there is a long list of references there are a number of relevant recent key references in this area of research that for unknown reasons are not cited. Please reconsider some as below.

Field et al. Nat Commun. 2018 Jan 9;9(1):116.

Shain et al. Nat Genet. 2019 Jul;51(7):1123-1130.

Karlsson et al. Nat Commun. 2020 Apr 20;11(1):1894.

Durante et al. Nat Commun. 2020 Jan 24;11(1):496.

Pelster et al J Clin Oncol. 2021 Feb 20;39(6):599-607.

Lutzky et al., Phase 2 Trial of Nivolumab plus Relatlimab in Metastatic Uveal Melanoma. (CA 224-294). SMR 2022.

Mariani et al. Br J Cancer. 2023 Sep;129(5):772-781.

Carvajal et al. Nat Rev Clin Oncol. 2023 Feb;20(2):99-115.

Thank you for suggesting these important references. We have added all of them except:

Lutzky et al., Phase 2 Trial of Nivolumab plus Relatlimab in Metastatic Uveal Melanoma. (CA 224-294). SMR 2022.

This was a plenary talk rather than a full paper and we are limited in terms of citation number. However, we do cite:

Tawbi, H. A. et al. Relatlimab and Nivolumab versus Nivolumab in Untreated Advanced Melanoma. *N Engl J Med* **386**, 24-34, doi:10.1056/NEJMoa2109970 (2022).

This also references anti-LAG3 treatment of metastatic UM.

Reviewer #5 (Remarks to the Author): expertise in immunology analysis

Despite the unprecedented therapeutic benefits of immunotherapies (esp. immune checkpoint blockade (ICB)) in patients with cutaneous melanoma (CM), patients with uveal melanoma (UM) have not been responding well to these novel immunotherapeutics, which has been attributed at least partially to UM being “cold”. In this manuscript, Leonard-Murali, et al. reported that UMs are actually not cold, at least in UM patients screened for adoptive cell therapy (ACT) with tumor-infiltrating T cells (TILs), as more than 50% of the metastases harvested in these patients have tumor-reactive TILs. Instead, the team reasons that the lack of response in UM patients to ICB and tebentafusp (a bispecific glycoprotein 100 peptide-HLA-directed CD3 T cell engager) is due to the extrinsic factors in UM that suppress the expansion of these reactive TILs. In support of this, the investigators showed that TILs isolated from UMs can be successfully expanded *ex vivo*, which, upon reinfusion back to the patients, can lead to appreciable tumor reduction. To predict which patients/metastases contain these TILs, the authors developed an immune metric called uveal melanoma immunogenomic score (UMIS), based on 2394 genes (1527 protein-coding genes and 867 non-coding/unclassified/pseudo genes) derived from Principal Component 2 (PC2). Subsequent evaluations of UMIS showed that metastases with UMIS <0.2 rarely produced reactive TIL cultures, indicating 0.2 is a preoperative threshold for surgical resection of “good and productive” metastases as sources of TIL preparation. Furthermore, UMIS above 0.246 predicted a significantly improved progression-free and overall survival of patients receiving adoptively transferred TILs. While these results support a threshold UMIS as a criterion in selecting metastases from UM patients for TIL expansion, some concerns remain:

1. While UMIS may be useful in identifying metastases for better TIL expansion, it is noteworthy to mention that most of the times it is not about which metastases to resect but rather about whether all the resectable metastases would provide enough materials for TIL expansion. Moreover, although UMIS inversely correlated with a reported immune resistance program (Fig. 4j), it did not predict/correlate ICB responses, nor did it predict a better overall survival.

Thank you for the insightful comments. In this study we have shown that UMIS not only predicts which metastases harbor TIL with anti-tumor specificity, but also the ability of these TIL to expand *ex vivo* (Fig. 5e,f,g,h,i,j). We did not associate UMIS with the amount of procured tissue. It is correct that UMIS did not correlate with ICB responses or better overall survival in this cohort (Extended Data Fig. 9a). However, the main finding of our study suggests that UMIS is a metric for occult tumor immunogenicity (Fig. 5f shown below) that is not potentiated by ICI or tebentafusp. However, upon *ex vivo* liberation, these quiescent tumor reactive TIL show dramatic expansion, suggesting that they can be enabled with TIL therapy.

2. While it is plausible to reason that extrinsic factors in UM suppress TIL expansion in vivo, these factors remain to be defined, a description of which would strengthen this study.

Thank you for this important observation. We attempted to define potential extrinsic factors with our single cell RNAseq analysis and did find that low UMIS tumor cells expressed higher levels of beta catenin (*CTNNB1*) than high UMIS tumor cells. This is a known immune exclusion program (Spranger, S., Bao, R. & Gajewski, T. F. Melanoma-intrinsic β -catenin signalling prevents anti-tumour immunity. *Nature* 523, 231-235 (2015). <https://doi.org/10.1038/nature14404>) that we further linked to expression of *SNHG7*, a non-coding RNA found to be associated with beta catenin expression in multiple cancers: (Chen, Y. et al. Knockdown of lncRNA SNHG7 inhibited cell proliferation and migration in bladder cancer through activating Wnt/ β -catenin pathway. *Pathol Res Pract* 215, 302-307 (2019). <https://doi.org/10.1016/j.prp.2018.11.015>; Bian, Z. et al. The role of long noncoding RNA SNHG7 in human cancers (Review). *Mol Clin Oncol* 13, 45 (2020). <https://doi.org/10.3892/mco.2020.2115>; Najafi, S. et al. Oncogenic Roles of Small Nucleolar RNA Host Gene 7 (SNHG7) Long Noncoding RNA in Human Cancers and Potentials. *Front Cell Dev Biol* 9, 809345 (2021). <https://doi.org/10.3389/fcell.2021.809345>; Ren, J. et al. Long noncoding RNA SNHG7 promotes the progression and growth of glioblastoma via inhibition of miR-5095. *Biochem Biophys Res Commun* 496, 712-718 (2018). <https://doi.org/10.1016/j.bbrc.2018.01.109>).

We have described these factors within the results and discussion (lines 443-461).

“Metastases with low UMIS (versus high UMIS) had a paucity of TIL and were composed of tumor cells with higher beta-catenin transcript expression

(CTNNB1), which has been described as a transcriptional repressor of BATF3-lineage dendritic cell recruitment of CD8+ T cells^{30,50,51}. In contrast, high UMIS metastases had lower tumor cell expression of CTNNB1, increased APC expression of T cell chemoattractant ligands (CXCL10 and CXCL9), greater tumor reactive TIL recruitment, and markedly elevated MHC expression on multiple cell populations within the tumor microenvironment, suggesting prominent interferon signaling. Thus, we believe Wnt/beta-catenin signaling to play an important role in promoting immune exclusion in metastatic UM, similar to prior reports in metastatic CM^{30,32,50-52}. Surprisingly, we identified only a single metastasis with a possible activating somatic mutation of the Wnt/beta-catenin pathway, suggesting hotspot mutations are not a common driver of beta-catenin overexpression in UM metastases⁵³. However, we did observe a strong correlation between the expression of the long non-coding RNA, SNHG7, and CTNNB1. Based upon several reports that SNHG7 is a positive regulator of CTNNB1 and compelling evidence that *in vitro* knockdown of SNHG7 leads to downregulation of the Wnt/beta-catenin pathway in various other cancers³⁴⁻³⁸ we are investigating the mechanistic role of this non-coding RNA in driving T cell exclusion in UM metastases and therapeutic strategies to potentially abrogate its effect in low UMIS metastases.”

3. Although UMIS was the strongest performer as both a correlative and classification metric for predicting ex vivo TIL reactivity, it remains to be determined whether it is really “significantly better” than other previously defined predictive biomarkers such as T cell GEP, Teff/IFN-g GEP, etc. This is important, given the complex composition of the 2394 genes used to calculate UMIS that would make it difficult to adopt in the clinic, particularly in other institutions.

Thank you for this comment. We agree that our ongoing prospective trial will be required to determine the true performance of UMIS. However, we do feel that the performance of UMIS is clearly superior to the other biomarkers in predicting TIL reactivity as both a categorical and continuous variable based upon the data that we present (Fig. 5g,h shown below).

Based upon our ongoing prospective use of UMIS, we believe that this testing could quickly be adapted to the clinic at other institutions. RNAseq is routinely performed for many other GEPs and our current bioinformatic pipeline is streamlined to provide UMIS output values in five to seven days using tissue from a core-needle biopsy.

4. Single random biopsies (i.e., central core fragments) from 100 metastases were utilized in this study to “facilitate a clinically relevant and minimally invasive biopsy approach for in situ tumor”. A potential concern with this approach is how representative these single biopsies are to whole tumors. Additionally, further elaboration is needed on how biopsy of central core fragments would be done in situ.

Thank you for this comment. We agree that sampling bias and tumor heterogeneity are obstacles for any biomarker, however, we were encouraged to find that UMIS (a single fragment-derived metric) could robustly predict the anti-tumor reactivity of *ex vivo* expanded TIL from the tumor in our cohorts. We acknowledge that UMIS may be less reliable in larger tumors or poorly viable metastases and have included this potential limitation in the discussion (lines 510-512).

“Further, although UMIS was found to correlate with TIL reactivity across 24 geographically unique tumor fragments, the predictive ability of UMIS in larger and more heterogenous lesions needs further study.”

With respect to clinical translation, we have found that *in situ* core-needle biopsy of patient metastases is sufficient to generate reliable transcriptomic data and produced comparable data to multiple spatially obtained tumor fragments. We have added language (lines 316-320) and data to support this observation (Extended Data Fig. 8a,b shown below).

“Additionally, we validated that UMIS remained consistent across spatially distinct areas of individual tumors (Extended Data Fig. 8a) and could also be obtained from minimally invasive core biopsies (Extended Data Fig. 8b).”

5. Detailed description of how principal components (particularly, PC2) were defined should be provided, as this is key to the calculation of UMIS. Similarly, every cluster (A, B, C, or D) has all of the 100 studied metastases. How were these clusters (A-D) “clustered”?

Thank you for this suggestion. Principal component analysis was performed using the top 10% most variable genes, and principal component gene and sample loadings were extracted from this model. Sample loadings provided the coordinates to map the 100 metastases along the principal components (Extended Data Fig. 2b,c). Correlation of sample loadings for individual principal components with sample enrichment scores for individual hallmark gene set collection pathways yielded Spearman rho values for each PC-gene set combination. We performed unsupervised hierarchical clustering of these values for PC1, PC2, and PC3 and all 50 hallmark gene set collection pathways (Fig. 2a). The natural clustering defined clusters A, B, C, and D. The gene loadings of PC2 were used to select genes used to eventually calculate UMIS.

This information is given in the results and methods sections (lines 96-101; lines 137-142).

“To determine whether specific cellular pathways and processes were associated with specific PCs, we correlated PC coordinates (1, 2, and 3) with enrichment scores for each of the canonical hallmark gene sets from the Human Molecular Signatures Database (Supplementary Table 3)¹⁸. Unsupervised clustering (Euclidean distance) of the PC-gene set correlations (Spearman’s rho) identified 4 discrete clusters (A, B, C, and D) with unique biologic motifs (Fig. 2a).”

“Rather than biasing this gene list with supervised filtering, we utilized the entire list of 2394 genes to facilitate discovery of novel biologic processes. Further, to enable single-sample prospective analysis, we employed a cohort-independent rank-based gene set scoring method (singscore¹⁸) to calculate enrichment scores for individual biopsies based upon transcript abundance (transcripts per million; TPM).”

6. Minor points: Line 29, “began focused studies” may read better with “focused our studies”; please clarify “~0” on Fig. 4i; please elaborate more on how autologous APCs were prepared (Fig. 5b).

Thank you for the clarifying language, this change has been adopted.

FDR values of ~0 are presented as such due to the lower limit of numeric processing in R (the program we used for all statistics). In R the lowest possible numeric value is roughly $1e-324$. Thus, we presented this as ~0 rather than listing arbitrary lower limit numbers. This information has been added to the methods (lines 968-969).

"In R the lowest possible numeric value is roughly $1e-324$. Thus, we presented values less than $1e-324$ as ~0 rather than listing arbitrary lower limit numbers."

Autologous APCs refer to patient specific monocytes, which were isolated from peripheral blood mononuclear cells using a CD14+ selection technique (Miltenyi Biotec). We have added this language and added the reagent to Supplementary Table 17 (lines 754-755).

"Monocytes were isolated from peripheral blood mononuclear cells using the CD14+ MicroBead isolation kit (Miltenyi Biotec)."

Reviewer #1 (Remarks to the Author):

All of my concerns have been addressed.

Reviewer #4 (Remarks to the Author):

Thank you for an extensive and valuable clarification of the data and the interpretation in the revised ms. I am especially happy with the data for additional patients and correlation of the findings to clinical outcomes. My overall impression is that the ms has been largely improved by adjusting and commenting to all the questions raised by the reviewers. I have nothing more to add. Congratulations to a nice executed study.

Reviewer #5 (Remarks to the Author):

The authors have successfully addressed all my concerns. I have no further questions.